

# Assessing Green and Blue Water: Understanding Interactions and Making Balance between Human and Nature

Ganquan Mao[1,2], Junguo Liu[1], Feng Han[1], Ying Meng[1], Yong Tian[1], Yi Zheng[1], Chunmiao Zheng[1]

[1]School of Environmental Science and Engineering, Southern University of Science and Technology, Shenzhen, 518055, China

[2] School of Water Resources and Hydropower Engineering, Wuhan University, Wuhan, 430072, China

*Correspondence to*: Junguo Liu (junguo.liu@gmail.com, liujg@sustc.edu.cn)

**Abstract.** Water resources assessment is crucial for human well-being and ecosystem's health. Assessments by considering both the blue and green water are of great significance as the green water plays a critical but often ignored role in the terrestrial ecosystem, especially in arid and semi-arid regions. Many approaches have been developed for green and blue water valuation, while few of them considers the interrelationship between them. This study proposed a new framework for green and blue water assessment by considering the interactions between them in an arid endorheic river basin where hydrological cycling is dramatically altered by human activities. Results show that even though the green water is the dominant water resources, the blue water is also critical. Most of the blue water are transformed to green water through physically and human induced processes to meet the water demand of ecosystems. Time and spatial variability of water supply and consumption forms totally different blue and green water regimes in different ecosystems. We also found that human use an increasing share of water with the decrease of the water availability. The massive water use by human reduces the water use for natural ecosystems. This indicates that natural ecosystems will take a higher risk of freshwater use when the water use competition increases. This study provides crucial information to better understand the interactions between green and blue water by assessing water resources in an explicit way. It also provides crucial implications for water management aiming to make the balance between humankind and nature.

## 1 Introduction

Renewable freshwater is a foundation for both terrestrial and aquatic ecosystems (Jackson et al., 2001). Sufficient water supply is essential for all organisms, including humans, for their survival (Oki and Kanae, 2006). Today, freshwater is becoming increasingly scarce in many regions of the world. Water scarcity has become a worldwide problem and its intensity has even increased due to climate change and human activities, e.g. unsustainable land management and water management (Liu et al., 2017; Veldkamp et al., 2017). This turns to a major constraint to our socio-economic development and a threat to livelihood in the world. As demand for water increases due to the economic development and population growth, there is an increased competition for water resources between agriculture, livestock, fisheries, forestry, energy and other sectors, with unpredictable impacts for livelihoods and environment (Johansson et al., 2016). Therefore, renewable



freshwater availability assessment is crucial for better water resources management for resilience against water scarcity under the changing environment (Rockström et al., 2009a).

Freshwater essentially stems from precipitation. Conceptually, fresh water resources can be divided into green water resources and blue water resources. The green water is the site-specific precipitation that does not run off but adds more or

less temporarily to soil water storage and eventually consumed by ecosystems through evapotranspiration, while the blue water is surface and ground water stored in rivers, lakes, aquifers and dams that can be extracted for human use (Falkenmark and Rockström, 2006). Early conventional water resources assessment and management considers only the blue water, while the green water has been ignored (Eastham et al., 2008; Shiklomanov, 2000; Xu and Singh, 2004). However, green water plays a critical role in terrestrial ecosystem, especially in arid and semi-arid regions (Liu et al., 2009a; Rockström et al., 2007;

Rost et al., 2008). Water resources assessments by considering both the blue and green water has become more and more diversified after the concept of green water was introduced by Falkenmark (1995). In particularly, after Falkenmark and Rockström (2006) conceptualized a wider green and blue water assessment approach for water resources planning and management, a great number of novel green and blue water researches have appeared. For instance, Schuol et al. (2008) combined SWAT model and GIS interface, together with SUFI-2 calibration procedure, successfully simulated the blue and

green water for the entire African continent at a detailed sub-basin level and monthly basis. Liu et al. (2009a) used the GEPIC (GIS-based EPIC) model to calculated the global consumptive green and blue water use for crop production and highlighted the importance of green water, especially for rain-fed agriculture. Zang et al. (2012) simulated the spatial and temporal distribution of blue and green water for an inland river basin in Northwest China, while the spatial variation and temporal trend of green water, blue water and green water coefficient are explicitly discussed. Lathuillière et al. (2016)

investigated the trade-offs between blue water resources and green water resources in the Amazon basin in light of future agricultural production and potential irrigation to assess costs and benefits to terrestrial ecosystems, in particular the land and biodiversity protection and regional precipitation recycling. In addition, many other studies have been done that are related to the green and blue water resources (Johansson et al., 2016; Mekonnen and Hoekstra, 2011; Rost et al., 2008; Sulser et al., 2010). These green and blue water related researches generally can be categorized into two groups depend on different

perspectives of the investigations: (1) Assessing green and blue water availability and their dynamics by using hydrological model or water-balance model. (2) Assessing green and blue water use or consumption by using the water resources model, agriculture model or dynamic vegetation model. Both strategies are important to provide scientific implications for water management, as the water resources from precipitation are recycled exclusively through water consumption in terms of evapotranspiration processes (Lathuillière et al., 2016).

Our earth has now entered in Anthropocene, a geological epoch that the Humanity's impact on the Earth is so profound (Crutzen, 2006; Van Loon et al., 2016), leading to increased pressures, competition and conflicts over freshwater resources between humankind and nature (Vörösmarty et al., 2000). Thus, green and blue water assessments become more and more urgent, especially in a coupled human-natural system, where both the green and blue water are critical for humankind and ecosystems (Johansson et al., 2016). Water resources availability and water consumption analyses are two main points of





entry for the state of the art of green and blue water assessments. Also, many studies have been done to investigate the climate and anthropogenic impacts on green and blue water (Chen et al., 2014; Liu, 2009; Liu et al., 2009b; Zang et al., 2015). However, few researches have paid attentions on the interconnection between green water and blue water.

Green and blue water are interlinked, the change of blue water drives also the green water changes and vice versa
(Rockström et al., 2009b). Several studies showed that human-driven changes in land cover, land use and water use alter the evaporation flux from terrestrial to atmosphere (Destouni et al., 2012; Gordon et al., 2005; Jaramillo et al., 2013; Wang and Hejazi, 2011), as well as the amount of moisture stored in the unsaturated soil layer as the soil moisture feeds the evaporation flux (Destouni and Verrot, 2014). Such green water related evaporation changes also affect runoff generation and therefore the consumptive use of blue water (Destouni et al., 2012; Jaramillo et al., 2013; Jaramillo and Destouni, 2014).
Thus, it is necessary to shift the strategy of traditional green and blue water resources assessment by also considering the interactions between them, i.e. beyond the water balance (McDonnell, 2017).

On the other hand, we are now in an era when more data become available for hydrological simulation at different scales (basin, continental, or even global scales). Although green and blue water availability can be sufficiently captured by the water balance accounting model which requires least datasets, compartmentalization of the parts of the terrestrial water cycle
could have a controlling influence on green and blue water storage dynamics in time and space (Cravotta et al., 2014). Various types of data allow us to drive our sophisticated models to depict the hydrological processes in an explicit way by tracking all the necessary water fluxes and storages in hydrological cycling. Thus, understanding of the interconnection between green water and blue water is not only more challenging, but also more inspiring, than just bulk quantification of water availability or water consumption in water resources assessments.

In this study, we investigated the green and blue water dynamics by considering the interactions between them in an arid endorheic river basin in China that is suffering strong anthropogenic impacts. In arid and semiarid areas, green water resources are critical and exchanges between green water and blue water are key hydrological processes and vital to sustainability of local society and ecosystem (Schyns et al., 2015). Human activities further alter the green and blue water interactions, imposing an additional stress on the water-limited environment. Understanding the complicated relationships
between green water and blue water is of great importance to the water resources management and ecological conservation in such areas.

This study aims to take into consideration of the exchanges between green water and blue water in water cycle, applying an integrated surface water and groundwater model for water resources assessment with two main objectives. (1) How integrated hydrological modeling could efficiently and effectively simulate green and blue water dynamics while
emphasizing the interlinkages between them; (2) How the implication of such green and blue water assessment could support basin-scale water resources management to address human-nature water conflicts.



## 2 Data and methods

### 2.1 Study area

The Heihe River Basin (HRB) is selected as the study area of this work and the reasons are: (1) HRB is located in the arid area and water in this area is limited; (2) HRB are impacted by heavy human activities and the hydrological cycling is dramatically altered; (3) HRB has strong groundwater and surface water exchanges which influences the interactions between green water and blue water.

The HRB is the second largest endorheic river basin in China with the majority located in Northwest China and a minor part in Mongolia. The Heihe River originates in the Qilian Mountains and discharges into the Juyanhai Lake, with a total basin area of 0.24 million km$^2$. Our research domain covers the middle and lower HRB plus a portion of the Badain Jaran Desert which hydraulically connected with HRB through its groundwater system (Figure 1). The upper HRB is excluded from the domain due to the reason that the upper HRB is still under a natural condition without human influences and it has sufficient water resources for the ecosystems. The total area of the domain is 90, 589 km$^2$. It consists of different types of land use/land cover (LULC) such as farmlands (5.6%), grasslands (5.9%), forests (1.1%), water bodies (1%), deserts (82.1%, including Gobi deserts), and others (4.2%). The farmlands are intensively irrigated, mainly for corn and winter-wheat planting.

**[Insert Figure 1]**

The climate and hydrological conditions and land use/land cover are slightly different for different parts in the domain. For instance, the middle HRB contains a large area of alluvial fans and floodplains, and has nearly all the intensively irrigated farmlands. The elevation of middle HRB ranges from 1200 to 3880 m. The annual average temperature here is about 8 °C, while the annual precipitation varies in space from 50 to 400 mm, with an average of 145 mm. The lower HRB is a vast Gobi-desert area and it is relatively flat with a mean elevation of about 1000 m. The vegetation mainly develops in the floodplains along the main river. The annual average temperature is about 10 °C and the annual precipitation amount here is extremely small (< 50 mm). Another big part of the domain is the Badain Jaran Desert that has many tall stationary dunes and numerous scattered lakes (probably fed by the groundwater) (Jiao et al., 2015). The annual average temperature is about 9 °C and the annual precipitation is around 110 mm. Since we are only interested in a part of the entire HRB, our study area is not physically closed. There are more than 30 perennial rivers, including the main Heihe River, bring around 3.5 billion cubic meters per year (m$^3$/year) of surface runoff from the Qilian Mountains (upper HRB) to the research domain.

### 2.2 Model

Usually, lumped or semi-distributed models are used for regional water resources assessment as the analysis is normally done on a basin or sub-basin scale (Xu and Singh, 2004). Here, a fully distributed hydrological model is selected for this research due to the two reasons below. (1) This study aims to assess the water resources by investigating the interlink between green water and blue water. The selected model is capable to simulate all the necessary hydrological elements for





this analysis due to the capacity of the model for detailed depiction of interactions between groundwater and surface water. (2) Gridded hydrological simulations from distributed model are essential for spatial investigation on green and blue water.

An improved GSFLOW (Coupled Ground-Water and Surface-Water Flow Model) was used in this study (Tian et al., 2015b), which is capable to explicitly consider irrigation, water diversion and groundwater pumping. GSFLOW integrates a

hydrological model PRMS (Precipitation-Runoff Modeling System) and a groundwater model MODFLOW (Modular Ground-Water Flow Model), which perform 2-D surface hydrology simulation and 3-D groundwater simulation, respectively (Markstrom et al., 2008). In the surface domain, hydrologic response units (HRUs) are the basic computing units, which can be either regular grids or irregular polygons, while the subsurface domain is discretized into finite difference girds. To simulate the interactions between surface water and ground water, a vadose zone between the soil zone and aquifer is defined

in GSFLOW, which is handled by the Unsaturated Zone Flow package (UZF1) (Niswonger et al., 2006) associated with MODFLOW. A "gravity reservoir" in the vadose zone is specified for each HRU as a storage in which the HRU exchanges water with the MODFLOW grid(s). The Streamflow Routing package (SFR2) (Niswonger and Prudic, 2005) and Lake package (LAK3) (Merritt and Konikow, 2000) are integrated in GSFLOW to simulate streams and lakes, respectively. In reaches where stream water is connected with groundwater, the stream-aquifer exchange is calculated based on the head

difference using Darcy's law (Darcy, 1856; Hubbert, 1957). More details about GSFLOW and its improvements can be found in Tian et al. (Tian et al., 2015a).

## 2.3 Data sources

The improved GSFLOW model has a rich description in ecological and hydrological processes, multisource data is desired for precise and thorough simulation. In this study, the data used in the modeling were obtained from the Heihe Program Data

Management Center (http://www.heihedata.org) and the details are listed in Table 1. The data used in this study are grouped into two categories. The first category is the data for model setup and initial parameterization, which includes Digital Elevation Model (DEM) (Farr and Kobrick, 2000), land use (Hu et al., 2015a, 2015b; Zhong et al., 2015), soil type (Dai et al., 2013; Wei et al., 2013), irrigation system (Hu et al., 2008), river network and so on. These data are essential for the model to define the topography, drainage system, boundary conditions, vertical segmentation of the subsurface and so on.

The second category of the data is dynamic model inputs that consists of model-derived climate forcing (Xiong and Yan, 2013), meteorological observation, surface water diversion and groundwater pumping information. These data are used to drive the model. This large amount of multisource data may differ in tempo-spatial scale and time span, an explicit description of the strategy for data pre-processing can be found in our previous work Tian et al. (Tian et al., 2015a).

**[Insert Table 1]**

## 30 2.4 Framework of the green and blue water assessment

In this study, we proposed a new framework for green and blue water assessment. The improved GSFLOW model is applied and is driven by multisource data to simulate necessary water fluxes and water storages for the assessment. Firstly, for the





application of hydrological simulation for the entire domain, uniform 1 km x 1 km grids were used for both the surface and subsurface domains in the model. The entire research domain contains 90,589 pixels, and each grid cell in the surface domain represents an individual HRU. The flow paths across HRUs is defined by using the Cascade Routing Tool (CRT) developed by Henson et al. (2013). Since the improved GSFLOW model has already been well calibrated and validated in

this region by Tian, Zheng, Zheng et al. (2015a) and Tian, Zheng, Wu et al. (2015b), it is directly run for the study area without any further parameter tuning. The model was run at a daily scale from 1 January 2000 to 31 December 2012, while the first year (2000) was treated as a "spin-up" period to eliminate the impact of initial conditions (mainly soil moisture) on the model simulations.

Figure 2 shows the framework for hydrological simulation of necessary variables. After the simulation, green and blue water

resources are calculated for assessments and the complete strategy consists of the following steps:

1. The green water and blue water resources are firstly calculated for each pixel and then summed up for analyses at different scales, e.g. the entire domain or different ecosystems.

2. The green water resources from precipitation are calculated by summing up the infiltration simulated by the model for a certain period (e.g. annual scale), as the infiltrated water from precipitation will be stored in the unsaturated

soil and eventually be used by the terrestrial ecosystems.

3. The blue water resources from precipitation are calculated by summing up the model simulated surface runoff, subsurface runoff and the groundwater recharge.

4. The transformation from blue water to green water consists of three parts, the capillary water, the canal seepage, and the irrigation. Both the capillary water and the canal seepage are simulated by the model, while the irrigation

water is an input variable.

**[Insert Figure 2]**

In addition to the strategy for green and blue water calculation, some basic terms are defined below:

- The water availability in this study refers to the amount of received water resources for a certain period.
- The water consumption in this study refers only to the evaporation (including also the interception) in the model

because the industry and domestic water use are not relevant for green- and blue water interactions. The green water consumption refers to the evaporation in terrestrial pixels and the blue water consumption refers to the evaporation in open water pixels.

The consumption ratio of human refers to the fraction of water use by human ecosystem, i.e. farmland.

**2.5 Green water coefficient**

Precipitation is partitioned into runoff and infiltration that recharges the soil when it reaches the surface (Rockström et al., 1999), which forms the blue and green water. The green water coefficient (GWC) is defined as the ratio of green water resources to the total water resources (green water + blue water) from precipitation and it can be calculated simply by the equation below (Liu et al., 2009a).



$$GWC = \frac{G_p}{B_p + G_p}$$

where $B_p$ and $G_p$ are blue and green water resources from precipitation, respectively. GWC reflects the precipitation partitioning ratio and is often used to analyze green and blue water from the perspective of water availability or water supply (Chen et al., 2014; Liu et al., 2009a; Liu and Yang, 2010; Zang et al., 2012).

## 3 Results and discussion

### 3.1 Domain averaged green and blue water flow chart

In order to investigate the green and blue water resources in an explicit way, a detailed green and blue water availability and consumption analysis have been done and the results are shown in the green and blue water flow chart (see Figure 3). The results are derived from hydrological simulations from 2001 to 2012 and are shown at an annual scale. The annual average precipitation in the research domain is 95.3 mm, which is 8.63 billion m$^3$ in volume. It is found that 86% (7.40 billion m$^3$/year) of the water from precipitation turns into green water resources that stored in soil and only 14% (1.23 billion m$^3$/year) of precipitation run off when it reaches the surface. In addition to the water resources from precipitation, there are also a large amount of water (3.93 billion m$^3$/year) from the upstream that charges the research domain, which accounted for 72.8% of total blue water and 30.7% of total water resources. The water resources are dominated by green water with a high GWC of 0.86, while the green water percentage of total water resources is 57.8% by considering the blue water replenishment from upstream. This means that the green water resources are the main water resources in this area, consistent with other studies for this region (Zang et al., 2015; Zang and Liu, 2013; Zuo et al., 2015).

**[Insert Figure 3]**

Our research domain is an endorheic river basin, therefore, there is no water flows out of the area and all the water entered in this area will eventually evaporated into the atmosphere. According to the definition in Falkenmark and Rockström (2006), the water evaporated directly through open water are accounted as the blue water consumption, which is 0.39 billion m$^3$/year and is only 3% of the total water consumption. The rest of the water resources (97%) are consumed by the terrestrial ecosystems and other land uses, e.g. bare soil and urban area, which are countered as green water consumption. This is due to the fact of limited open water area in an arid or semi-arid region (Sánchez-Carrillo et al., 2004). Actually, at the global scale, about two-thirds of the water resources are consumed as green water consumption (Gerten et al., 2005; Rost et al., 2008). The average annual total green water consumption for the study area is 12.41 billion m$^3$/year and the consumption for each ecosystem, i.e. farmland, forest, grassland and desert, are 3.03 billion m$^3$/year, 0.22 billion m$^3$/year, 1.77 billion m$^3$/year and 5.99 billion m$^3$/year, respectively. The desert ecosystem has the highest green water consumption (48.3% of the total green water consumption) due to its large area, while the second highest green water consumption ecosystem is farmland (24.4%) partly due to the intensive irrigation (Ge et al., 2013). In addition, about 1.40 billion m$^3$ (11.2%) green water is consumed each year by other land uses including bare soil and urban area. The total water consumption (12.80 billion





m$^3$/year) is higher than the total water availability (12.56 billion m$^3$/year), as about 0.24 billion m$^3$/year (2.7 mm in depth) groundwater is abstracted each year. It is important to point out that the total green water consumption (12.41 billion m$^3$/year) is 67% higher than the original green water storage (7.4 billion m$^3$/year). Which means a large amount of additional water resources is needed for this region and transformed into green water resources to meet the consumption.

For the entire domain, there are 5.40 billion m$^3$/year blue water in total, while only 7.2% (0.39 billion m$^3$/year) of them are evaporated from open water and the rest are transformed into green water which are eventually consumed by ecosystems. The blue water to green water transformations are mainly driven by three factors, i.e. irrigation, canal seepage and capillary uptake, due to human activities and hydrological processes. There are about 3.63 billion m$^3$ blue water used each year for irrigation through water pumping from wells (0.82 billion m$^3$) and water diversion from rivers (2.81 billion m$^3$). Due to the

irrigation systems, 0.83 billion m$^3$/year blue water are transformed additionally to green water through the canal seepage. In addition, there are also 0.55 billion m$^3$ blue water recharge the soil moisture through the capillary rise. All these blue to green water transformations are critical for ecosystems, especially the agriculture ecosystem, as the water demand are extremely higher than the water stored in the root zone layer (green water from precipitation).

Our study area covers only the middle and lower HRB, although the water resources from precipitation is still the main water

resources for the study area, additional water from upstream (upper HRB) is also crucial for the ecosystems in this region. This means that the runoff generation in the upstream could have pronounced impacts on the ecosystems in the middle and lower HRB. The imbalance between water availability and water consumption causes the groundwater depletion with an average rate of 0.24 billion m$^3$/year. This would provide implications for the water management in this region as the groundwater depletion could cause serious problems for both human and ecosystems in the future. Furthermore, the

inequality between green water storage and green water consumption implies abundant transformation from blue water to green water. This detailed water flow chart analysis helps us to further understand the eco-hydrological processes beyond the water balance and provides crucial information for water management.

## 3.2 Analysis of spatial and temporal variability of the water resources

In addition to assess the water resources at basin scale, it is also necessary to reveal their spatial and temporal patterns, which

provides information in a more detailed way. Firstly, the spatial patterns of critical water elements are investigated and shown in Figure 4, including total water resources from precipitation, green water resources from precipitation, irrigation and total water consumption. It is important to state here that in the model, green water resources from precipitation only refers to the water from precipitation that stored in the soil. The water runs through the soil and then recharges the groundwater or routed into channel is excluded from this part.

All of these four variables are shown at an annual scale at the spatial resolution of 1 km, it is easy to see that all of them vary in space. Figure 4 (a) shows the averaged total water resources it can receive per year for the entire domain, which is the annual precipitation in this region. It shows very large variability while southeast of the domain can receive more water resources than other area. In the north of the study area, the received water resources are quite low. In some region, the





precipitation is even less than 50 mm/year. Figure 4 (b) shows the green water resource that can be received for the area. It follows exactly the pattern of precipitation without doubt due to the fact that it comes from the precipitation. Also, a clear river network is shown in the map because the precipitation that directly reaches the channel is calculated as the blue water instead of the green water. The blue water map is not show here, as the blue water can be simply calculated by using total

water resources from precipitation subtract green water resources from precipitation.

Figure 4 (a) and (b) show water resources including total water resources and green water resources from precipitation, while (c) shows the human induced water allocation, i.e. irrigation and (d) shows the total water consumption, i.e. evaporation. Figure 4 (c) and (d) show similar patterns since the irrigation impacts directly the evaporation. Also, differences can be found in the southeast part of the domain, which is caused by heavy precipitation in this region. By comparing the maps of

precipitation (Figure 4 (a)) and irrigation (Figure 4 (c)), results show that irrigation is the dominated water resources for most of the irrigated area. In some of the region, the magnitude of irrigation is even larger than that of precipitation. The differences between precipitation map (Figure 4 (a)) and evaporation map (Figure 4 (d)), especially the large difference of the range, show the large imbalance between water resources received from precipitation and water consumption. This reflects the fact that the upstream inflow, the blue water from outside of the domain, is crucial for the region to meet the

water requirement. It also emphasizes the importance of transformation from blue water to green water, which support the ecosystems and bridges the gaps between water supply and water use spatially.

**[Insert Figure 4]**

The temporal variability of total water availability and total water consumption has also been analyzed and shown in Figure 5. Here, different colors in blue indicates blue waters from different sources. The dark blue indicates the blue water from

upstream, i.e. inflow, while the light blue implies the blue water from precipitation. It can be seen that both the total water resources (includes the green water, blue water and the inflow from upstream) and the total water consumption (evapotranspiration) varies in time, while the green water is still the main water resources for each year with larger band width in the figure. The sum of green and blue water represents the total precipitation. With a visual inspection, a clear pattern shows that the annual variability of green water depends very much on the temporal distribution of precipitation for

this region. This is in line with findings of previous research (Zang et al., 2012). Moreover, the green water resources have stronger temporal variability (Coefficient of variation: 30.2%) than the blue water resources from the precipitation (Coefficient of variation: 20.8%). A statistical analysis also shows a steady annual pattern for inflow with only 9% of the Coefficient of variation. The water consumption follows the similar annual pattern of the total water resources. This is due to the reason that the water availability is main constraint for consumption in this region (Elliott et al., 2014). However, there

are still inequality between total water resources and total water consumption, while this imbalance reflects the total water storage change of the entire domain.

**[Insert Figure 5]**





### 3.3 Explicit green and blue water analysis for different ecosystems

Different ecosystems have different hydrological processing mechanisms, thus the philosophy of water use for different ecosystems might also be different (Gao et al., 2017; Savenije and Hrachowitz, 2017). In addition, the spatial and temporal heterogeneity which is shown in previous analyses will also effect the water use by different ecosystems. Therefore, it is

necessary to investigate the blue and green water regimes in different ecosystems. In this section, the green and blue water assessment is done for different ecosystems. Here, four major ecosystems are included, i.e. farmland, forest, grassland and desert, and explicit investigations are shown in Figure 6. Different to the gross water flow chart investigation (Figure 3) that reflects the total amount of different flows, here this figure shows the water flows in depth (unit: mm/year). This allows us to see the intensity of water fluxes and their exchanges and also provides a clear impression by representing the amount with

corresponding arrow size.

**[Insert Figure 6]**

Regarding to the water resources, our investigation shows that the grassland ecosystem received the highest amount of precipitation per unit area (173.9 mm/year), which is very close to the precipitation received by farmland (169.8 mm/year). The annual precipitation for forest is 100.9 mm/year and for desert the amount is only 77.4 mm/year in an average.

Precipitation partitioning is the critical process for green and blue water formation and the partitioning ratio is also different for different ecosystems due to the reason that the land cover has significant influence for the runoff generation (Hernandez et al., 2000; Sriwongsitanon and Taesombat, 2011). The GWC for farmland, forest, grassland and desert are 0.82, 0.71, 0.79 and 96.9, respectively. This means that the forest ecosystem has highest runoff coefficient and nearly all the precipitations fall in desert are store in the soil rather than runoff. The blue water consists of surface runoff, subsurface runoff and

groundwater recharge and the percentage of each components are different due to the different mechanisms of hydrologic cycling for ecosystems (Savenije and Hrachowitz, 2017).

The water consumption for these four ecosystems are quite different varying from 91.5 mm/year in desert ecosystem to 756 mm/year in farmland ecosystem. Although the desert consumed most of the water resources in this region (see Figure 3), the consumption per unit area is however low (91.5 mm/year). The farmland ecosystem has the highest water consumption per

unit area of 756 mm/year partly due to the large amount of irrigation. Our analysis shows that except desert, all the other three ecosystems received additional water resources through irrigation with different magnitude. The farmland received irrigation of 442.7 mm/year per unit, more than three times of the green water stored in the soil. This follows the nature of arid regions that the water is limited for agriculture which requires a lot more water through irrigation (Al-Zu'bi, 2007). The forest also received 171.5 mm/year irrigated water and the irrigation for grassland is only 36.2 mm/year. Even though the

unit-averaged irrigation for grassland is lower than that for forest, the received amount of irrigation for grassland is larger due to the larger area of grassland compared to forest (see Section 2.1). Regarding to the volume of irrigated water, grassland received 7.4% of the total irrigation (0.26 billion m³/year) and forest obtained 6.6% of total irrigation (0.24 billion m³/year), while farmland took 86% of the irrigation (3.13 billion m³/year). Moreover, because of the irrigation system, there are quite





amount of water are leaked from the irrigation canal, i.e. the canal seepage. This part of the water is also an important source for ecosystems as the leaked water recharges the soil and can be used by the plants. This kind of soil moisture recharge including the irrigation is different to the soil moisture recharge from precipitation, as the water is reallocated from the stream to soil (Scott et al., 2000). Here, it is referred to the transformation from blue water to green water. It can be seen that

the amount of canal seepage depends very much on the magnitude of irrigation and it also reflects the water use efficiency of irrigation (Wang et al., 2015). In addition to the human induced blue to green water transformation, irrigation and canal seepage, there is also physically induced transformation, e.g. capillary uptake, which draw the water from groundwater and provide considerable amount of water for ecosystems (Zhang, 2016). The capillary uptake in forest is 124.8 mm/year and is even higher than the soil water recharge from the precipitation. The capillary for farmland, grassland and desert are much

less and are 64.6 mm/year, 53.1 mm/year and 16.5 mm/year, respectively. This is because of the root of the plant in forest could reach deep soil and the transpiration could cause higher moisture deficit pressure than other ecosystems (Zhu et al., 2009).

### 3.4 Water consumption dynamics between human and nature

The HRB is located in the arid region, where the water resources are limited for both the human and nature ecosystems.

However, agriculture is important for local economy in HRB, which requires quite a lot of water (Cheng et al., 2014). To support the farmland ecosystem, i.e. the human ecosystem, considerable amount of blue water is needed to meet massive water consumption through irrigation (see Section 3.1 and 3.3). However, the use of the water by human will undoubtedly narrow the water use for natural ecosystems, such as forest ecosystem, grassland ecosystem and desert ecosystem. In order to balance the water consumption between human and nature, a holistic understanding on water consumption dynamics for both

human and nature is necessary.

In this section, the relationship between human water consumption and nature water use are investigated and the results are shown in Figure 7. The water consumption ratios are calculated for both human and nature, while only the results for human are shown in the plots as the consumption ratio of nature equals to one minus the consumption ratio of human. Here, we only consider the agriculture water use as the water use for human, since the sum of domestic and industry water use is less than 5%

of the total human water use in the entire HRB (Cheng et al., 2014; Li et al., 2015; Wang et al., 2009). Moreover, the domestic and industry water use is irrelevant to green water and blue water interactions, since almost all of the return water of domestic and industry water abstraction flows back into the river channel through pipes causing no exchanges between green and blue water (Vanham et al., 2018). Indeed, the neglect of domestic and industry water use will influence the total amount of the water consumption calculation. However, it will only slightly affect the results due to the small amount and

we are more interested in the ecosystem water consumption on the perspective of hydrology that can reflect the interlinkages of green and blue water. Therefore, the water consumption for forest, grassland and desert are considered as the natural water use, and the water consumption for farmland are calculated as the human water use.

**[Insert Figure 7]**



With a visual inspection of Figure 7, both the blue water consumption ratio and total water consumption ratio of human varies in different years. The human consumption ratio of blue water ranges from 39.4% in the year of 2007 to 61.5% in the year of 2001, while the human consumption ratio of total water resources falls between 22.1% in the year of 2002 and 32.2% in the year of 2005. A linear regression is applied (Figure 7: solid line in plots) to show the linear relationship between the water availability and human water consumption ratio. The human consumption ratio of blue water is relatively higher than that of total water and both the blue water consumption ratio and total water consumption ratio of human show a clear decreasing trend with the increase of the corresponding water availability. This is due to the reason that blue water use can be controlled by human through water allocation. Water resources are used by human with a higher priority compared to the nature as the water consumed by human ecosystem, i.e. farmland, will benefit the local economy. In dry years, e.g. the year of 2004, nearly 60% of the blue water resources are used by human ecosystem. Since the green water resources for crops are extremely insufficient and additional water resources are required. In wet years, e.g. the year of 2007, water stress in low in the domain and few blue water resources might already meet the water demand of crops. In this case, more blue water resources will be left for natural ecosystems and the blue water consumption ratio of human will be low (39.4%).

Our results also reveal that the natural ecosystems will face increased risk over water consumption (especially the blue water consumption), as the human consumption ratio arise if the water availability decreases. In other words, the increasing of water competition between human and nature will impose restrictions on natural water use. This analysis helps us better understanding the competition relationship between water uses by human and nature in changing conditions and provides crucial implications for water management that aims to balance the water use between human and nature.

## 4 Conclusion

In this study, we have explicitly assessed the green and blue water resources by using a new framework and considering the interactions between green and blue water in the Heihe River Basin, the second largest endorheic river basin in China. An integrated hydrological model is applied to simulate complex eco-hydrological processes and provide essential hydrological simulations for this assessment. Major findings of this study include the following.

- Even though the green water resources are the major resources in the research area - an arid river basin, the blue water resources from upstream are also crucial for the ecosystems in this region to meet the water demand. The transformation from blue water to green water plays a key role in the completed water cycling in this area as the water required for evaporation are extremely higher than the water stored in the root zone area (green water from precipitation).

- Both the water availability and water consumption vary in time and space in the research area. Different hydrological processing mechanisms in ecosystems together with the spatial and temporal heterogeneity of water supply forms totally different green and blue water regimes in different ecosystems. The farmland ecosystem highly relies on the irrigation, while the forest relies on both the irrigation and capillary water. Both the grassland and





desert ecosystems mainly rely on the green water from precipitation, while the desert ecosystem almost generates no runoffs.

- • The historical relationship between human water use and nature water use indicates that the blue water resources are used by human with a higher priority. Water consumption ratio of human increases with the decrease of the water availability. The natural ecosystems may take a higher pressure when the water competition between human and nature increases.

This study for the first time assesses the water resources by considering not only the blue and green water but also their interconnections. By taking the advantages of a sophisticated hydrological model, this study helps us to better understand the interactions between green and blue water, especially for different ecosystems. It allows us to explicitly assess the green and blue water resources beyond the water balance, while the traditional methods using lumped or semi-distributed model might be insufficient. This study also investigated the blue and green water from both water supply and water consumption perspectives, while conventional studies focus only on one of them. Such sophisticated research framework allows us to take into consideration of all the important factors into water resources assessment as possible. The detailed analyses of green and blue water dynamics bring us a step further to understand the human and nature water use dynamics. This study highlights the need to consider the interactions between green and blue water for water resources assessment due to the strong exchange between green and blue water, especially in the area with strong human activities. It provides essential implications for water management under the changing environment that aims to make the balance between humankind and nature and towards sustainable development.

However, for this study there are a few shortcomings. First, the current work omitted the industry and domestic water uses due to the lack of data. Even though it did not much influence the interactions between green and blue water, the calculation of water consumption for human is slightly affected thus causing a small impact on the investigation of water consumption dynamics between human and nature. Second, the results are simulated with one model. Although the model has been calibrated and validated in several previous studies in the same region (Li et al., 2017, 2018, Tian et al., 2015a, 2015b), simulations may be constrained by the fundamental assumption and approaches used in this model. Third, this study is only a fundamental investigation on water resources. It shows the natural ecosystems may take a higher pressure when the water competition between human and nature increases, which provides implications for water management under the changing environment. Further research is needed in the future to quantify the potential risk each grid cell or different ecosystems may take. Thus, the hotspot area that may suffer higher risk on water use can be identified. This would make the research more practical and meaningful.

## Acknowledgment

This study was supported by the National Natural Science Foundation of China project (91325303, 41625001, 41571022, 51711520317) and National Key Research and Development Program in China (2017YFA0603704). Additional support was





provided by the Southern University of Science and Technology (G01296001). This work was partially funded by Shenzhen Municipal Science and Technology Innovation Committee through project Shenzhen Key Laboratory of Soil and Groundwater Pollution Control (ZDSY20150831141712549). All the data used in this work are available at the data center of the "Integrated research on the eco-hydrological process of the Heihe River Basin" (http://www.heihedata.org) and the
World Data System Cold and Arid Regions Science Data Center at Lanzhou (http://card.westgis. ac.cn).

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

**Tables**

**Table 1 Data sources for hydrological simulation**

| Category | Data | Time of Data | Spatial resolution |
| --- | --- | --- | --- |
| Model setups | DEM | 2000 | 90 m × 90 m |
| | Land use | 2000, 2007, 2011 | 1:100,000 |
| | Soil texture | 2012 | 1 km × 1 km |
| | Normalized difference vegetation index | 2000–2012 (every 10 days) | 1 km × 1 km |
| | River network | 2000 | 1:100,000 |
| | Irrigation system | 2006 | 1:100,000 |
| | Hydrogeology map | 2002 | 1:500,000 |
| | Borehole data | Multiple time spots | 257 locations |
| | Boundary river inflow | 2000–2012 (daily/monthly) | 15 stations |
| | Boundary groundwater inflow | 2000–2012 (yearly) | Boundary grids |
| Model inputs | Model-derived climate data | 2000–2012 (six hours) | 3 km × 3 km |
| | Meteorological observations | 2000–2012 (daily) | 19 stations |
| | Surface water diversion | 2000–2012 (monthly) | 46 districts |
| | Groundwater pumping | 2000–2012 (yearly) | 46 districts |





# Figures

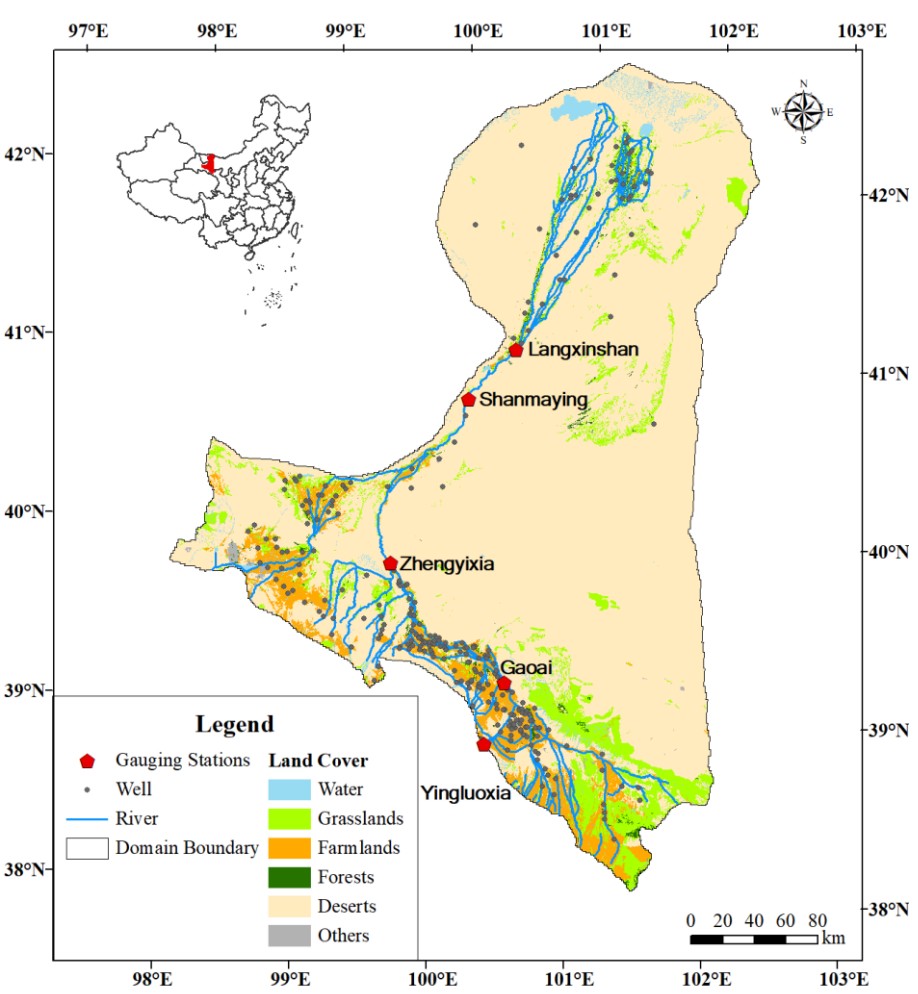

**Figure 1: The study domain.**





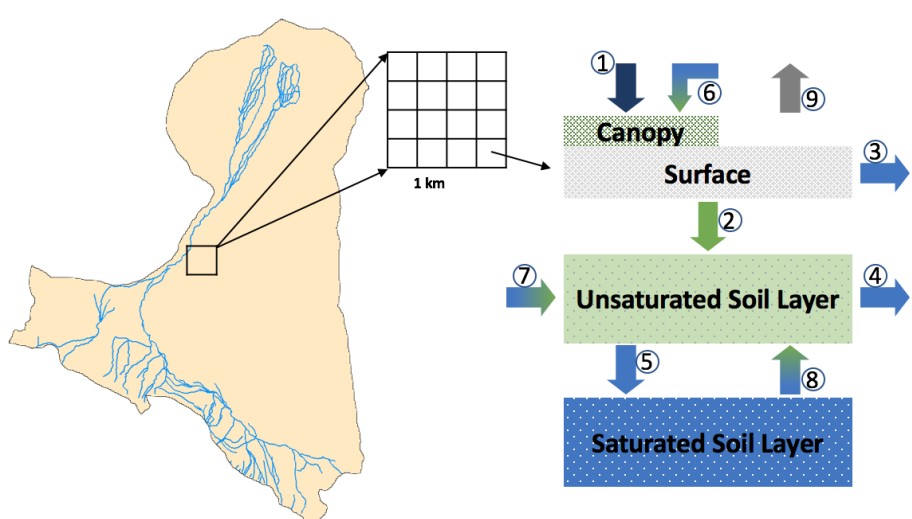

| 1. Precipitation | 2. Infiltration | 3. Surface runoff |
| 4. Subsurface runoff | 5. Groundwater recharge | 6. Irrigation |
| 7. Canal seepage | 8. Capillary uptake | 9. Evaporation |

**Figure 2: The framework for hydrological simulation in the research domain.**




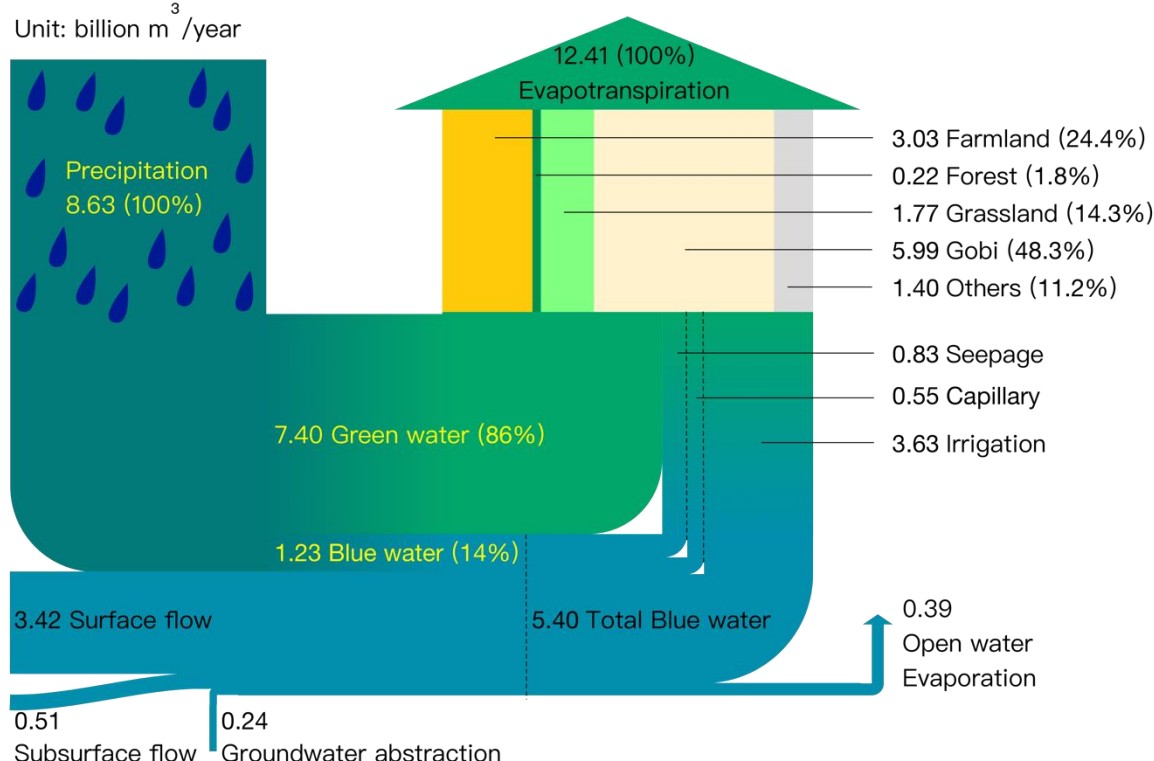

**Figure 3: The green and blue water flow chart for the entire domain at an annual scale. Left part shows the water resources availability of the domain which consists of water from precipitation, surface and subsurface flow from upstream and groundwater abstraction. The big vertical arrow in the right side shows the water consumption by different ecosystems (Land Use) and the small vertical arrow shows the water consumption on open water region. Blue colors indicate the blue water flows and green colors stands for green water flows. A mixer color of blue and green implies the precipitation, while a gradient transition from blue to green shows the transformation of blue water to green water. Numbers in parentheses indicate the percent of total, while the yellow text stands for water resources from precipitation and the black text represents water consumption. The width of the band reflects the quantitative of each element.**





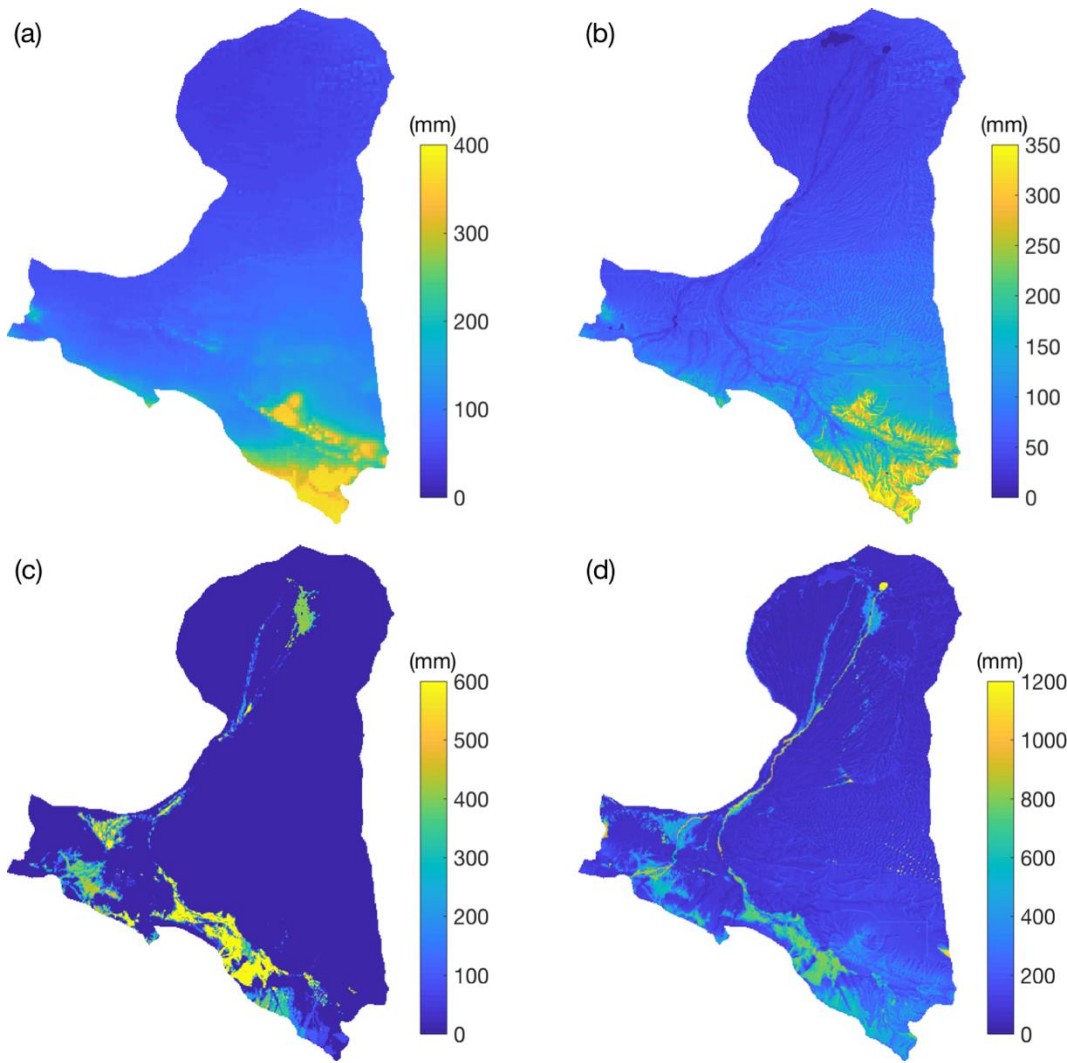

**Figure 4: Spatial analysis of water supply and water consumption including (a) total water resources from precipitation, (b) green water resources from precipitation, (c) irrigation and (d) total water consumption in the research domain. The results are shown at an annual scale with the average mean based on the data from 2001 to 2010. The color axis is scaled differently based on the different ranges of the data.**




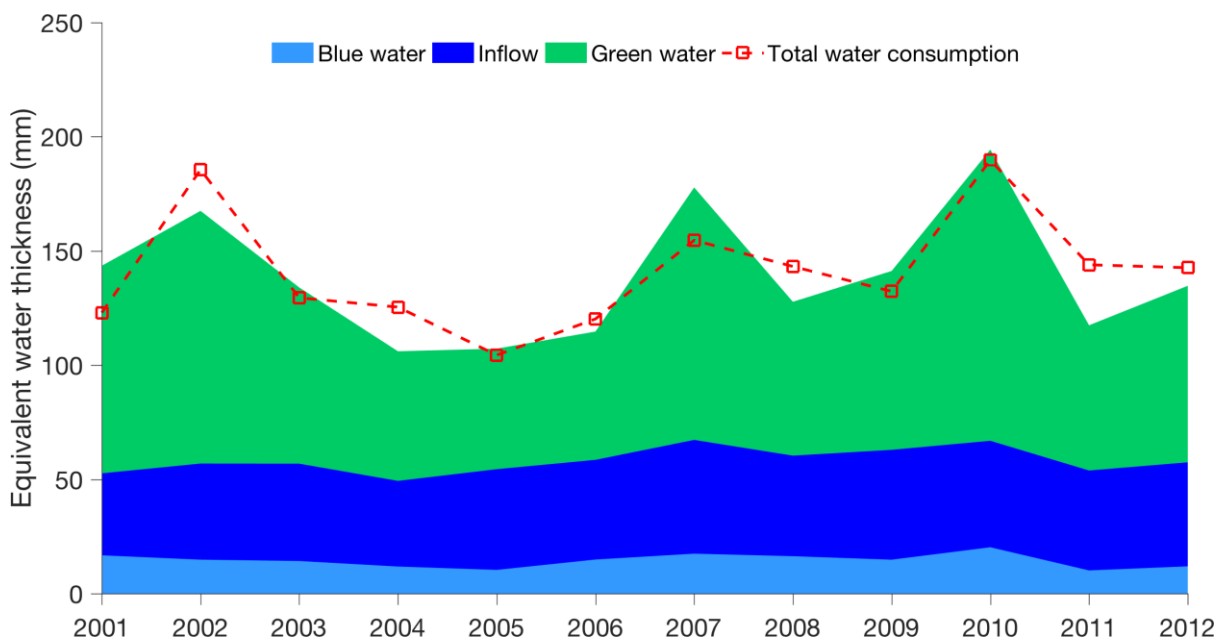

**Figure 5: Temporal distribution of the water resources and water consumptions. The data is summed up from all the pixels for the entire domain and then converted into equivalent water thickness for the sake of convenience for comparison. The light blue indicates the blue water from precipitation and the green color represents the green water from precipitation, while the dark blue stands for the blue water from upstream, i.e. the inflow of the domain. The red dashed line shows the water consumption through evapotranspiration.**





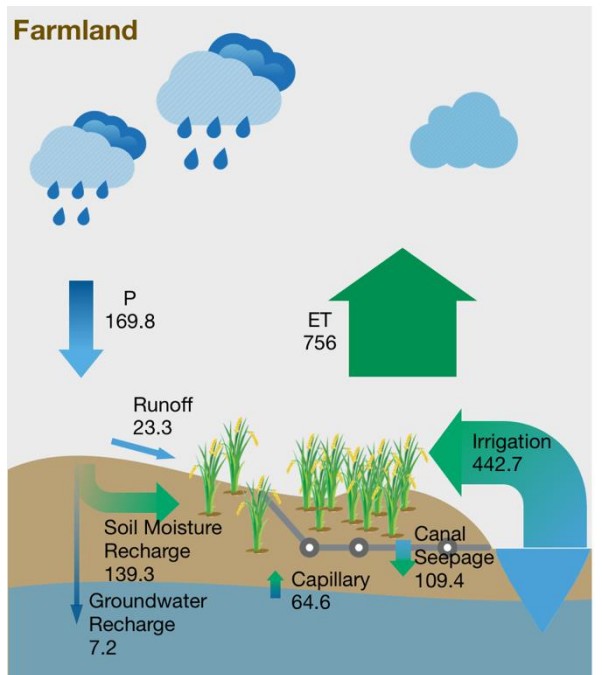

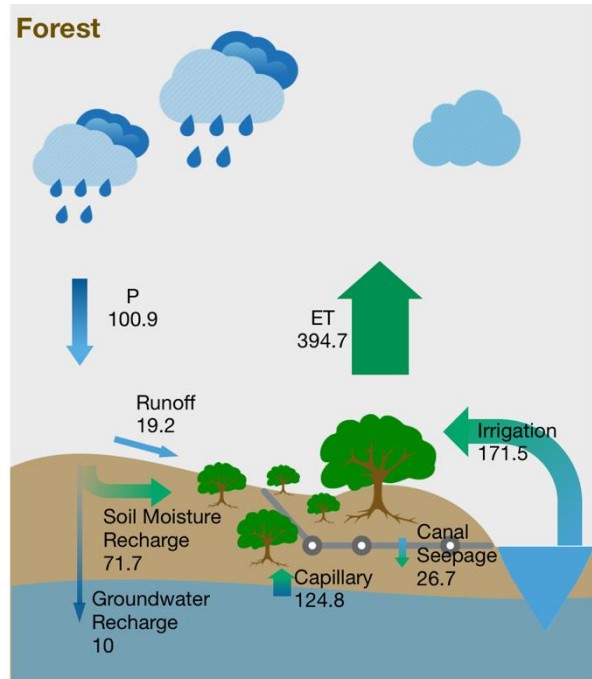

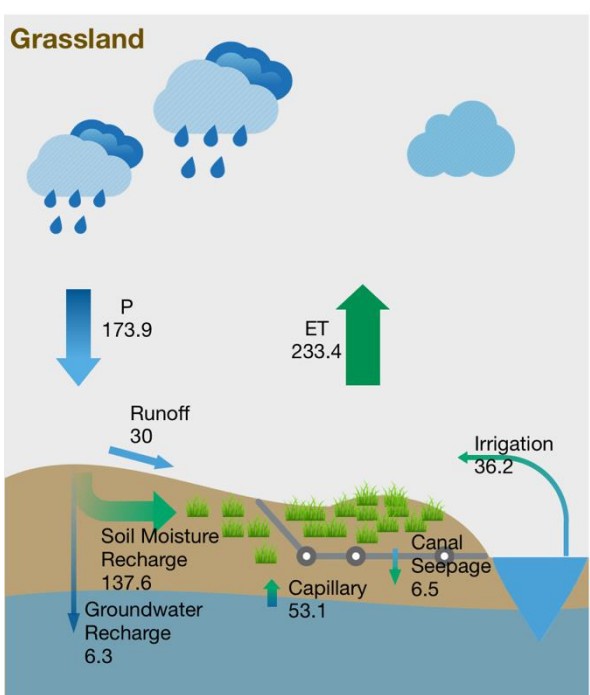

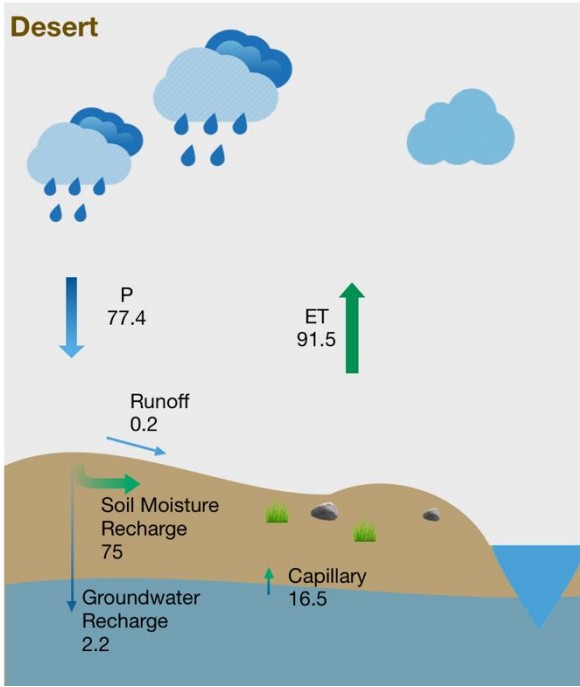

**Figure 6: Explicit green and blue water assessment for different ecosystems at an annual scale. The data is summed up from all the pixels that from a certain ecosystem and then converted into equivalent water thickness. The unit is mm/year for all of these four plots. Blue arrows indicate the blue water flows and green arrows stands for green water flows. Arrows with a gradient transition from blue to green stands for the transformation of blue water to green water. The size of the arrow reflects the magnitude of water flows.**




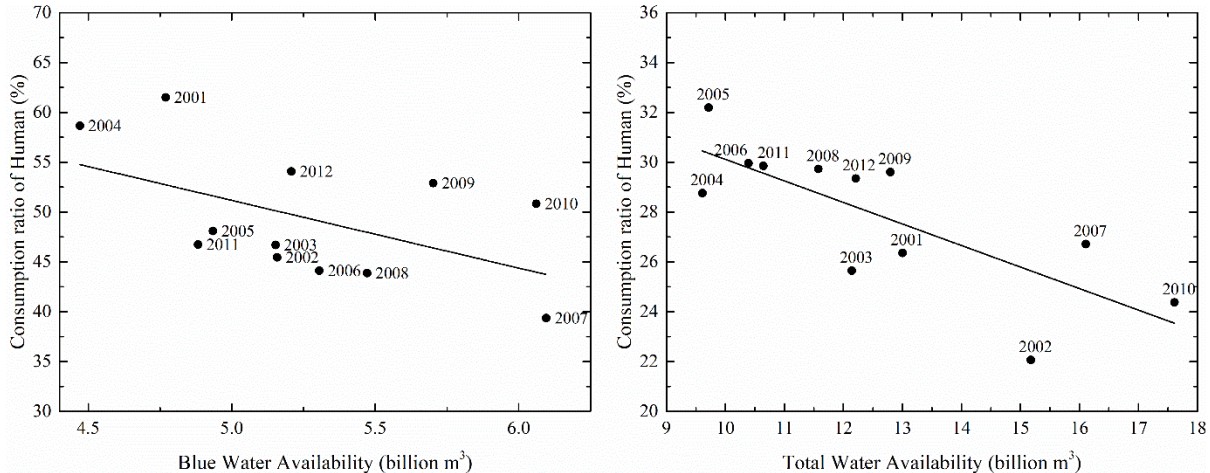

**Figure 7: The relationship between consumption ratio by human and water availability. The left plot shows the results based on blue water availability and blue water consumption ratio, while the right plot shows the results based on total water availability and total water consumption ratio. The black dots represent the consumption ratio of human in different years. The solid line stands for the linear regression of the data. All the results are calculated at annual scale based on the data from 2001 to 2010.**