# Peer review of "Assessing Green and Blue Water: Understanding Interactions and Making Balance between Human and Nature"

_Hydrology and Earth System Sciences, 2018_

## Short Comment (SC1) · 7 May 2018

1) In the current version of the manuscript, the distinction between semi-distributed and fully distributed hydrological models is not apparent. What is meant by semi-distributed hydrological model? What is meant by fully distributed hydrological model? On what basis the selected model is considered as a fully distributed hydrological model (see P-4 LN-28)? Is the definition of fully distributed model meant for simulating the "necessary" hydrological element for the analysis of interest (see P-4 LN-31)? If the definition of fully distributed model is preserved only for the "necessary" elements, how would it lead to conclude that the selected model is a fully distributed hydrological model?

[Figure]

Are there elements other than the necessary elements that are not fully described in the selected model? From the reader's point of view, having a fully distributed hydrological model is precluded with our understanding on the processes that define the hydrological system, even though many efforts have been made and are in progress to better understand the hydrological system. Moreover, from the reader's point of view, in no way, the selected model can represent a fully distributed hydrological model, considering the way the selected model has been conceptualized as described in the manuscript(see P-5 LN-1-15).

2) The current version of the manuscript is based on a model that was calibrated and validated by Tian et al, 2015(https://www.sciencedirect.com/science/article/pii/S1364815214003016). Therefore, the research work that is carried out and presented in the current version of the manuscript solely depends on the calibrated and validated model. Moreover, around 82.1% of the LULC (see P-4 LN-14) of the domain of interest is desert. Therefore, it becomes paramount to evaluate the calibrated and validated model. Otherwise, the evaluation of this manuscript will be based on the assumption that the authors of that manuscript (https://www.sciencedirect.com/science/article/pii/S1364815214003016) are reliable and well-known in the field of hydrology, and/or the editorial board went through the calibrated and validated model carefully and ensured that the model is flawless. Moreover, the published manuscript (https://www.sciencedirect.com/science/article/pii/S1364815214003016)) is not feely available for a reader of an open-access journal (i.e., HESS) to evaluate this manuscript.

3) As per the "new" framework (see P-1 LN-11) implemented by the authors, the green and blue water resources are calculated for each pixel and then summed up (see P-6 LN-11). What is meant by this framework (fully distributed? new framework?)? How did the authors implement the streamflow routing? How did the authors account the pixels that represent the stream network? How did the authors calculate the surface
runoff/excess rainfall at the uppermost point of a reach, which got routed along the reach? Moreover, what is the meaning of green water in the deserts (i.e., 82.1% of LULC)? Are these green water available in the root system for the plants in the deserts? If this is the case, the classification of LULC is misleading? Why are those areas considered as deserts?

4) As per the authors, the study area (i.e., HRB) is impacted by "heavy human activities" and the hydrological cycle is "dramatically" altered. Moreover, the study area has "strong" GW and SW exchanges (see P-4 LN-5). Do these statements need supportive texts/references?
* * *

---

## Short Comment (SC2) · 16 May 2018

The authors of this paper aim to improve the understanding of the hydrological interactions between green and blue water, and the relation between water for agriculture versus water for natural ecosystems. They study an arid endorheic river basin in China and use a coupled groundwater-surface water model. I have five major concerns with this manuscript:

1) The literature review on the one hand is very lengthy – going into many directions that seem not so relevant for this paper – while on the other hand key references are not included or not discussed properly. In the end it remains vague what the exact

contribution of this paper is and where the novelty lies.

Perhaps the most relevant study which is not mentioned is the one by Weiskel et al. (2014). They used a detailed distributed water balance model to simulate green-blue water fluxes across the US and develop a classification of hydrological regimes/units based on these green-blue water fluxes. Another study that looked into the interaction between green and blue water fluxes is the one by Chukalla et al. (2015). They developed a method to separate cropland evaporation into green and blue fractions based on the ingoing and outgoing fluxes of the water balance. Other recent studies of relevance that could be included are: Lathuillière et al. (2018) and Xu & Wu (2018). The paper by Schyns et al. (2015) is referred to in an odd context in the manuscript (page 3, lines 21-23).

It is not clear what is the (novel) contribution of this manuscript. The conclusion section contains some claims on the novelty of the research which are strongly overstated: (a) "This study for the first time assesses the water resources by considering not only the blue and green water but also their interconnections."; and (b) "This study also investigated the blue and green water from both water supply and water consumption perspectives, while conventional studies focus only on one of them". Regarding statement (a), the studies by Weiskel et al. (2014) and Chukalla et al. (2015) considered this in a detailed manner. Basically, all studies that use a hydrological model or vegetation or crop growth model with a proper water balance in there take into account the interaction between green and blue water (e.g. Rost et al. (2008); Hanasaki et al. (2010)). Regarding statement (b), papers on combined green-blue water scarcity have studied green and blue water consumption versus green and blue water availability (Rockström et al., 2009; Gerten et al., 2011; Kummu et al., 2014). See also Schyns et al. (2015).

2) The definitions of green and blue water in this manuscript deviate from previous studies for unknown reasons, and the definitions are mutually inconsistent. Since the focus of the manuscript is on the interactions between green and blue water flows (as put forward prominently in the title and introduction), this is a serious methodological

flaw, which really makes me question the scientific quality of this work.

Various definitions of green water exist (see Schyns et al. (2015) section 2.3), though most studies define the green water flow as actual evapotranspiration (or more preferably called evaporation (Savenije, 2004)) from land, excluding the part of evaporation that is the result of blue water resources that have been redirected to the soil moisture through irrigation, capillary rise, or natural flooding. The authors have chosen their own definition of green water: "The green water resources from precipitation are calculated by summing up the infiltration simulated by the model for a certain period (e.g. annual scale), as the infiltrated water from precipitation will be stored in the unsaturated soil and eventually be used by the terrestrial ecosystems." This definition is incomplete and inconsistent with how blue water is defined. Water that infiltrates into the unsaturated zone of the soil will in part evaporate – through soil evaporation and through plant transpiration – and in part it will add to groundwater and surface water through percolation and interflow. Rockström and Falkenmark (2000) refer to this as the 'second partitioning point'. It is thus not true that the "infiltrated water from precipitation will be stored in the unsaturated soil and eventually be used by the terrestrial ecosystems" as the authors state. Infiltrated water will in part contribute to blue water resources. Furthermore, the authors' definition of green water does not include the intercepted rainwater that evaporates. Evaporation of intercepted rainwater is also part of the green water flow, albeit a non-productive vapour flow (Rockström and Falkenmark, 2000).

The authors have the following definition of blue water: "The blue water resources from precipitation are calculated by summing up the model simulated surface runoff, subsurface runoff and the groundwater recharge." Since infiltrated precipitation contributes to subsurface runoff and groundwater recharge as explained above, the used definitions of green and blue water are inconsistent and double-counting occurs.

The authors speak of irrigation and capillary rise as a transformation of blue to green water, and use the following definitions of green and blue water consumption: "The green water consumption refers to the evaporation in terrestrial pixels and the blue

water consumption refers to the evaporation in open water pixels." I find this highly confusing, since it suggests that irrigation (and capillary rise) is accounted for by the authors as green water consumption, while previous studies all see this as blue water consumption (e.g. Oki & Kanae (2006); Rost et al. (2008); Liu & Yang (2009); Hoekstra & Mekonnen (2012); Hanasaki et al. (2010); Siebert and Döll (2010)). In fact, this means that this manuscript treats all agricultural water use as green water. An example from the paper: "...while the second highest green water consumption ecosystem is farmland (24.4%) partly due to the intensive irrigation" (page 7, lines 29-30). Moreover, the quoted definition suggests that open water evaporation is a form of blue water consumption, while in fact open water evaporation is purely natural, unless we are talking about open water evaporation of man-made reservoirs (Hogeboom et al. (2018)).

The definition of water availability that is put forward is also not clear: "The water availability in this study refers to the amount of received water resources for a certain period." What is meant by the received water resources? Simply precipitation?

3) The second objective of this manuscript is to study the relation between water for humans versus water for nature, as put forward in the introduction and the manuscript title. However, this is only addressed superficially without even mentioning the term 'environmental flow (requirements)' in the manuscript.

4) None of the three major findings presented in the conclusions are new insights. Three major findings are presented in the conclusions. The first one basically says that irrigation is important, since in arid areas soil moisture stemming from precipitation is insufficient for agriculture. The second one confirms this and mentions that the green-blue partitioning depends on the land use. The third one says that natural ecosystems may be under pressure when human water demand increases, and when water availability decreases the ratio of water use to availability increases (if demand remains the same). I fail to see what is new about these insights.

5) The overall writing style is not on par with the level of a high quality paper, as indicated by the above examples of overstatements and definitions that are not fully clear. Also many (vague) claims are made without proper justification. Some examples from the conclusions section: "It allows us to explicitly assess the green and blue water resources beyond the water balance, while the traditional methods using lumped or semi-distributed model might be insufficient."; "Such sophisticated research framework allows us to take into consideration of all the important factors into water resources assessment as possible."; "The detailed analyses of green and blue water dynamics bring us a step further to understand the human and nature water use dynamics."; "It provides essential implications for water management under the changing environment that aims to make the balance between humankind and nature and towards sustainable development."

Additional references mentioned in this comment:

Gerten, D., Heinke, J., Hoff, H., Biemans, H., Fader, M. & Waha, K. (2011) Global water availability and requirements for future food production, Journal of Hydrometeorology, 12(5): 885-899.

Hanasaki, N., Inuzuka, T., Kanae, S. & Oki, T. (2010) An estimation of global virtual water flow and sources of water withdrawal for major crops and livestock products using a global hydrological model, Journal of Hydrology, 384(3-4): 232-244.

Hoekstra, A.Y. and Mekonnen, M.M. (2012) The water footprint of humanity, Proceedings of the National Academy of Sciences, 109(9): 3232−3237.

Hogeboom, R.J., Knook, L. & Hoekstra, A.Y. (2018) The blue water footprint of the world's artificial reservoirs for hydroelectricity, irrigation, residential and industrial water supply, flood protection, fishing and recreation, Advances in Water Resources, 113: 285-294.

Kummu, M., Gerten, D., Heinke, J., Konzmann, M. & Varis, O. (2014) Climate-driven interannual variability of water scarcity in food production potential: a global analysis,

Hydrology and Earth System Sciences, 18(2): 447-461.

Lathuillière M.J., Coe, M.T., Castanho, A., Graesser, J. & Johnson, M.S. (2018) Evaluating water use for agricultural intensification in Southern Amazonia using the water footprint sustainability assessment, Water, 10(4): 349.

Rockström, J. & Falkenmark, M. (2000) Semiarid Crop Production from a Hydrological Perspective: Gap between Potential and Actual Yields, Critical Reviews in Plant Sciences, 19:4, 319-346.

Savenije, H.H.G. (2004) The importance of interception and why we should delete the term evapotranspiration from our vocabulary, Hydrological Processes, 18(8): 1507-1511.

Siebert, S. & Döll, P. (2010) Quantifying blue and green virtual water contents in global crop production as well as potential production losses without irrigation, Journal of Hydrology, 384(3-4): 198-217.

Weiskel, P.K., Wolock, D.M., Zarriello, P.J., Vogel, R.M., Levin, S.B. & Lent, R.M. (2014) Hydroclimatic regimes: a distributed water-balance framework for hydrologic assessment, classification, and management, Hydrology and Earth System Sciences, 18(10): 3855-3872.

Xu, H. & Wu, M. (2018) A first estimation of county-based green water availability and its implications for agriculture and bioenergy production in the United States, Water, 10(2): 148.

---

## Referee Comment (RC1) · HHG Savenije (Referee) · 22 May 2018

In this paper, the authors analyzed output from the GSFLOW model in the middle and lower Heihe River Basin (HRB), and investigated the division between green and blue water and their interaction in different ecosystems. There are quite a number of models that have been applied to the HRB, among which the GSFLOW model, which has been crosschecked with multi-source datasets.

Figures 3 and 6 provide a clear representation of the water balance of the middle and lower HRB and in different ecosystems. However, I am not sure if Figure 6 fully represents reality. It is assumed that all the precipitation that does not runoff goes

through the soil moisture stock. I doubt this. Apparently it is assumed that this water is transported through the root zone, and evaporated by transpiration. This cannot be true. Judging by the very low precipitation (about 170 mm/a), I would expect most of the precipitation on the lower Heihe to be partitioned into interception. In forested areas, the preferential infiltration may be substantial, but in forests interception is also substantial. On the desert, the infiltration is probably zero. Also on farmland and grassland, most of the precipitation will be captured by (leaf and ground) interception. I think the authors should make an effort to identify the partitioning of ET in Transpiration, Interception and Soil Evaporation.

**Minor Comments:**

I think that the last paragraph of the conclusion would be better placed in the discussion section.

I think that one publication on the Heihe is missing, which is the paper by Gao et al. (2014), which studied the runoff of the Upper Heihe river basin and providing the input to the middle HRB, see: Gao, H., Hrachowitz, M., Fenicia, F., Gharari, S., and Savenije, H. H. G.: Testing the realism of a topography-driven model (flex-topo) in the nested catchments of the upper heihe, china, Hydrology and Earth System Sciences, 18, 1895-1915, 10.5194/hess-18-1895-2014, 2014.

**Some small corrections:**

page 5 line8, do you mean by "grids"

page 5 line18, what is improved in the GSFLOW model?

page 8 line 3, "Which means...", please merge the two sentences.

page 8 line 33, in some regions, ...

page 9 line 4, "The blue water map is not shown here..."

page 9 line 15, "which supports the ecosystems and bridges the gap between..."

page 9 line 29, "water availability is the main..."

page 10 Section 3.3 what do you mean by "explicit"?

page 10 line 18, "all the precipitation", precipitation is uncountable.

page 10 line 29. "the forest also received 171.5 mm/year irrigated water..." this is very interesting conclusion. Please clarify how forest is irrigated. Is this really happening in the entire basin or only in urban and agricultural areas?

page 10 line 33. "there are quite amount of water are ..." this sentence should be rephrased.

page 13, line 7 it is better to say "for the first time...interconnections in the HRB".

page 13, line 10 "beyond the water balance". I am not quite convinced with this conclusion. It seems to be still in the framework of water balance.

---

## Referee Comment (RC2) · Anonymous Referee #2 · 6 Jun 2018

This paper analyses results of a previously applied hydrological model (gsflow) to assess green and blue water partitioning in the Heine basin in China. The paper is well written however unclear in his contribution and as a result it is difficult to judge if methods are tailored to the objectives or the objectives are a tailored to available results from previous studies. More specific comments follow:

1 The objectives of the paper state: "(1) How integrated hydrological modeling could efficiently and effectively simulate green and blue water dynamics while emphasizing the interlinkages between them; (2) How the implication of such green and blue water assessment could support basin-scale water resources management to address

human-nature water conflicts." Both objectives are too generic. First, surely there are studies that have addressed these problems. The objectives of the paper need to reflect what this study does in addition. Second, this paper does not address these objectives in general. It is limited to a specific application in the Heine catchment. This needs to be reflected.

2 Title is too generic. It could be a good title for a book or for a conference session. It would be appropriate if the paper was so comprehensive to address a problem that was never addressed and to a level that the problem is definitely solved. It is not the case of an application of a model on the Heihe basin.

3 The introduction speaks about geological eras and problems of mankind but should be tailored to the specific advances that the paper wants to make.

4 The model selection is a crucial methodological choice. A fully distributed hydrological model is selected for the following reasons: "(1) This study aims to assess the water resources by investigating the interlink between green water and blue water. The selected model is capable to simulate all the necessary hydrological elements for this analysis due to the capacity of the model for detailed depiction of interactions between groundwater and surface water. (2) Gridded hydrological simulations from distributed model are essential for spatial investigation on green and blue water." Both arguments are not convincing. Re argument 1, what are all the necessary hydrological elements needed to assess green and blue water components? I struggle to see why one would need to model everything if interested only in such partitioning. Re argument 2, why gridded simulations are essential? Can one go with a couple of HRUs?

5 The calibration and validation of the model is deferred to another study. This limits the relative contribution of this paper to an analysis of already available results, and complicates the assessment of the appropriateness of methods.

6 There is no uncertainty analysis in this paper and no comparison with alternative models. Considering that green and blue water are derived quantities form a calibration

on other type of data (ie streamflow), I suspect that there is large uncertainty in the analysis. Perhaps something that is difficult to do with such a complex model, but perhaps also something that would speak in favour of different methodological choices? It would be also good to see how different would results be if applying a much simpler model, and if there are arguments to prefer the results of one model over the other.

7 The treatment of the interception process is unclear. It appears that intercepted water has been incorporated in green water, but as already noted by other reviewers, this may be inappropriate. This comes back to clarifying what are the necessary processes to estimate green and blue water components (see point 4).

8 Most conclusions are not related to the paper objectives

---

## Author Comment (AC1) · 26 Jul 2018

We would like to thank Referee Prof. HHG Savenije for his interest in this topic and for the valuable comments to improve our manuscript. Based on the comments some recalculations have been performed. Our point-by-point response to the comments is given in the following:

**General comments from referee**

In this paper, the authors analyzed output from the GSFLOW model in the middle and lower Heihe River Basin (HRB), and investigated the division between green and blue water and their interaction in different ecosystems. There are quite a number of models that have been applied to the HRB, among which the GSFLOW model, which has been crosschecked with multi-source datasets.

**Comment 1**

Figures 3 and 6 provide a clear representation of the water balance of the middle and lower HRB and in different ecosystems. However, I am not sure if Figure 6 fully represents reality. It is assumed that all the precipitation that does not runoff goes through the soil moisture stock. I doubt this. Apparently it is assumed that this water is transported through the root zone, and evaporated by transpiration. This cannot be true. Judging by the very low precipitation (about 170 mm/a), I would expect most of the precipitation on the lower Heihe to be partitioned into interception. In forested areas, the preferential infiltration may be substantial, but in forests interception is also substantial. On the desert, the infiltration is probably zero. Also on farmland and grassland, most of the precipitation will be captured by (leaf and ground) interception. I think the authors should make an effort to identify the partitioning of ET in Transpiration, Interception and Soil Evaporation.

**Author's response**

Thanks for the comments. Indeed, we would like to use Figure 3 and Figure 6 to represent clearly the water balance, especially reveal the exchanges between blue and green water.

In Figure 6, the green arrow which is marked as the "Soil Moisture Recharge" is actually the part that consists of soil moisture recharge and interception. We are very sorry for this mistake which causing confusion here. We thank you for pointing it out. We have now corrected it. Since the interception, transpiration and soil evaporation are all accounted as the green water, thus we combined them together in this study as the evapotranspiration that represents the total green water flux.

We totally agree that the ET partition is important, especially for the moisture cycling. As different evaporation components plays different role in the hydrological cycle (van der Ent et al., 2014). However, it is out of the scope of this work. In this study, we focused on more the water resources assessment rather than the hydrological cycle. Therefore, we are more interested in the green and blue water flowing and the interconection between them rather than the partition of hydrological fluxes. To provide a clear representation of the movement of green and blue water without making the results unnecessary complex. We combined some fluxes on purpose, e.g. the surface runoff and subsurface runoff are combined as runoff. Even though the surface runoff and subsurface runoff has different mechanism on the perspective of hydrological processes. They play similar roles in green and blue water flow chart on the perspective of water resources assessment. Interception, transpiration and soil evaporation are orginally combined on the same purpose as runoff.

However, we still believe that the separation of canopy interception is necessary and could help improving the description of green and blue water flow regimes. As the canopy interception actually does not go in the soil and will directly evaporate back to the atmosphere from leaves. We have extracted the variable of canopy interception and updated the manuscript and the figures. The updated figure is shown in the following. We also inserted the following content in the manuscript "Eventhough interception is accounted as green water, it is still separated from total evporation. As it is directly evaporate back to the atmosphere from leaves and does not go into the soil. The interception for farmland, forest, grassland and desert are 47.5 mm/year, 30.8 mm/year, 41.5 mm/year and 0.6 mm/year, respectively. The irrigation is not intercepted due to the reason that flood and furrow irrigation are still the main irrigation modes in HRB during our research period (Zhou et al., 2015). There is also interception in desert area, since some regions are covered by desert vegetation that is considered in our model."

Currently, the model used in this study has only the module for canopy interception simulation. Therefore, the floor interception is not considered during the simulation. Also, in the model, the soil evaporation and transpiration are not separated inherently limited by the model structure. However, it will not affect the results of the green water flow regime, as they are all accounted as green water and play similar roles on the perspective of flow regime analysis. Furthermore, the irrigation is not intercepted due to the reason that flood and furrow irrigation are still the main irrigation modes in HRB during our research period (Zhou et al., 2015).

**Author's changes in manuscript**

We have change all the term of "evapotranspiration" that may cause confusion in the manuscript to "total evaporation" and stated in Page 7, Line 20 "The total evaporation consists of interception and evapotranspiration."

Page 10, Line 17.

We have inserted the following sentences before "The GWC for farmland ...".

Inserted sentences "Eventhough interception is accounted as green water, it is still separated from total evporation. As it is directly evaporate back to the atmosphere from leaves and does not go into the soil. The interception for farmland, forest, grassland and desert are 47.5 mm/year, 30.8 mm/year, 41.5 mm/year and 0.6 mm/year, respectively. The irrigation is not intercepted due to the reason that flood and furrow irrigation are still the main irrigation modes in HRB during our research period (Zhou et al., 2015). There is also interception in desert area, since some regions are covered by desert vegetation that is considered in our model."

**Minor Comments**

I think that the last paragraph of the conclusion would be better placed in the discussion section.

I think that one publication on the Heihe is missing, which is the paper by Gao et al. (2014), which studied the runoff of the Upper Heihe river basin and providing the input to the middle HRB, see: Gao, H., Hrachowitz, M., Fenicia, F., Gharari, S., and Savenije, H. H. G.: Testing the realism of a topography-driven model (flex-topo) in the nested catchments of the upper heihe, china, Hydrology and Earth System Sciences, 18, 1895-1915, 10.5194/hess-18-1895-2014, 2014.

[Figure]

[Figure]

[Figure]

[Figure]

**updated Figure 6**. Explicit green and blue water assessment for different ecosystems at an annual scale. The data is summed up from all the pixels that from a certain ecosystem and then converted into equivalent water thickness. The unit is mm/year for all of these four plots. Blue arrows indicate the blue water flows and green arrows stands for green water flows. Arrows with a gradient transition from blue to green stands for the transformation of blue water to green water. The size of the arrow reflects the magnitude of water flows.

**Answer**

Thank you for the suggestion. We have moved the last paragraph of the conclusions into the end of discussion section. '

Thank you for the nice paper which is related to the runoff generation in Upper Heihe river basin that provding streamflow for middle and lower HRB. We have cited the suggested relevant paper in our work in Page 8, Line 15.

**Author's changes in manuscript**

Page 13, Line 19.

The complete paragraph "However, for this study there are a few shortcomings. First, ..." has been deleted.

Page 12, Line 18.

A paragraph is inserted after the sentence "... that aims to balance the water use between human and nature."

Inserted paragraph "However, for this study there are a few shortcomings. First, the current work omitted the industry and domestic water uses due to the lack of data. Even though it did not much influence the interactions between green and blue water, the calculation of water consumption for human is slightly affected thus causing a small impact on the investigation of water consumption dynamics between human and nature. Second, the results are simulated with one model. Although the model has been calibrated and validated in several previous studies in the same region (Li et al., 2017, 2018, Tian et al., 2015a, 2015b), simulations may be constrained by the fundamental assumption and approaches used in this model. Third, this study is only a fundamental investigation on water resources. It shows the natural ecosystems may take a higher pressure when the water competition between human and nature increases, which provides implications for water management under the changing environment. Further research is needed in the future to quantify the potential risk each grid cell or different ecosystems may take. Thus, the hotspot area that may suffer higher risk on water use can be identified. This would make the research more practical and meaningful."

Page 8, Line 15.

A paper is cited here in the end of the sentence. "... additional water from upstream (upper HRB) is also crucial for the ecosystems in this region (Gao et al., 2014)."

**Some small corrections:**

page 5 line8, do you mean by "grids"

We are not sure if the question is "what do you mean by grids?", If so, the answer would be: "grids here is more or less like the pixels, in this study it represents the hydrologic response units (HRU) in the model"

page 5 line18, what is improved in the GSFLOW model?

One of the key improvement is the explicit consideration of irrigation, water diversion and groundwater pumping (Page 5, Line 4 in the manuscript).

page 8 line 3, "Which means...", please merge the two sentences.

Thank you for the suggestion, we have modified the sentence as following.

**original:** "It is important to point out that the total green water consumption (12.41 billion $m^3$/year) is 67% higher than the original green water storage (7.4 billion $m^3$/year). Which means a large amount of additional water resources is needed for this region and transformed into green water resources to meet the consumption."

**updated:** "It is important to point out that the total green water consumption (12.41 billion $m^3$/year) is 67% higher than the original green water storage (7.4 billion $m^3$/year), implying that a large amount of additional water resources is needed for this region and transformed into green water resources to meet the consumption."

page 8 line 33, in some regions, ...

It has been modified.

**original:** "In some region, ..."

**updated:** "In some regions, ..."

page 9 line 4, "The blue water map is not shown here..."

It has been modified.

**original:** "The blue water map is not show here..."

**updated:** "The blue water map is not shown here..."

page 9 line 15, "which supports the ecosystems and bridges the gap between..."

It has been modified.

**original:** "which support the ecosystems and bridges the gaps between..."

**updated:** "which supports the ecosystems and bridges the gap between..."

page 9 line 29, "water availability is the main..."

It has been modified.

**original:** "water availability is main..."

**updated:** "water availability is the main..."

page 10 Section 3.3 what do you mean by "explicit"?

"explicit" here refers to a detailed and clear investigation on green and blue water. To avoid the confusion, we have updated the section 3.3 to "Green and blue water analysis for different ecosystems"

page 10 line 18, "all the precipitation", precipitation is uncountable.

It has been modified.

**original:** "... and nearly all the precipitations fall in desert are store in the soil rather than runoff."

**updated:** "... and precipitations fall in desert are mainly stored in the soil rather than runoff."

page 10 line 29. "the forest also received 171.5 mm/year irrigated water..." this is very interesting conclusion. Please clarify how forest is irrigated. Is this really happening in the entire basin or only in urban and agricultural areas?

In our study, irrigation also happens in the forest area in the middle and lower HRB due to two reasons. First, Chinese government launched water use policy since the late 1990s to protect the environment in the middle and lower HRB, especially for conserving populus euphratica, a typical plant in HRB. (Sun et al., 2016; Zhang et al., 2015). Thus, a part of water in HRB will be used to irrigate the forest (Nian et al., 2014). Second, there are also forests located in the irrigation districts in middle HRB (Li et al., 2018). They received the irrigated water resources from the irrigation system.

We have insert this clarification of irrigation in forest into the manuscript in Page 10, Line 31 after the sentence "... of grassland compared to forest (see Section 2.1)".

page 10 line 33. "there are quite amount of water are ..." this sentence should be rephrased.

It has been modified.

**original:** "Moreover, because of the irrigation system, there are quite amount of water are leaked from the irrigation canal, i.e. the canal seepage."

**updated:** "Moreover, there are quite amount of water are leaked from the irrigation canal, i.e. the canal seepage."

page 13, line 7 it is better to say "for the first time...interconnections in the HRB".

Thank you for the suggestion. It has been changed accordingly.

**original:** "This study for the first time assesses the water resources by considering not only the blue and green water but also their interconnections."

**updated:** "This study for the first time assesses the water resources by considering not only the blue and green water but also their interconnections in the HRB."

page 13, line 10 "beyond the water balance". I am not quite convinced with this conclusion. It seems to be still in the framework of water balance.

Thank you for the comment. We agree with the referee and change the statement accordingly to "beyond the simple flow in and out framework." The supportive information for this conclusion is followed.

In this work, in addition to analysis the flow of green and blue water in and out of the HRB, i.e. the simple flow in and out framework, we revealed the interconnection between the green and blue water. In our research domain, the received green and blue water and consumed green and blue water are not balanced. We investigated not only how much of green and blue water flow in and flow out of the domain, but also how green and blue water transformed between each other to match the imbalanced green and blue water flow regime.

**References**

van der Ent, R. J., Wang-Erlandsson, L., Keys, P. W. and Savenije, H. H. G.: Contrasting roles of interception and transpiration in the hydrological cycle - Part 2: Moisture recycling, Earth Syst. Dyn., 5(2), 471–489, 2014.

Zhou, Q., Wu, F. and Zhang, Q.: Is irrigation water price an effective leverage for water management? An empirical study in the middle reaches of the Heihe River basin, Phys. Chem. Earth, 89–90, 25–32, doi:10.1016/j.pce.2015.09.002, 2015.

Sun, T., Wang, J., Huang, Q. and Li, Y.: Assessment of Water Rights and Irrigation Pricing Reforms in Heihe River Basin in China, Water, 8(8), 333, doi:10.3390/w8080333, 2016.

Zhang, A., Zheng, C., Wang, S. and Yao, Y.: Analysis of streamflow variations in the Heihe River Basin, northwest China: Trends, abrupt changes, driving factors and ecological influences, J. Hydrol. Reg. Stud., 3, 106–124, doi:10.1016/j.ejrh.2014.10.005, 2015.

Li, X., Cheng, G., Ge, Y., Li, H., Han, F., Hu, X., Tian, W., Tian, Y., Pan, X., Nian, Y., Zhang, Y., Ran, Y., Zheng, Y., Gao, B., Yang, D., Zheng, C., Wang, X., Liu, S. and Cai, X.: Hydrological Cycle in the Heihe River Basin and Its Implication for Water Resource Management in Endorheic Basins, J. Geophys. Res. Atmos., n/a-n/a, doi:10.1002/2017JD027889, 2018.

Nian, Y., Li, X., Zhou, J. and Hu, X.: Impact of land use change on water resource allocation in the middle reaches of the Heihe River Basin in northwestern China, J. Arid Land, 6(3), 273–286, doi:10.1007/s40333-013-0209-4, 2014.

---

## Author Comment (AC2) · 26 Jul 2018

We would like to thank Anonymous Referee # 2 for his/her interest in this topic and for the valuable comments to improve our manuscript. Our point-by-point response to the comments is given in the following:

**General comments from referee**

This paper analyses results of a previously applied hydrological model (gsflow) to assess green and blue water partitioning in the Heine basin in China. The paper is well written however unclear in his contribution and as a result it is difficult to judge if methods are tailored to the objectives or the objectives are a tailored to available results from previous studies. More specific comments follow:

**Author's response**

Thank you for the comment. The main contribution of our work includes the following. We did a thorough green and blue water assessment by investigating the blue and green water from both water supply and water consumption perspectives and considering the interaction between green and blue water, while conventional studies usually focus only on either water supply or water consumption and ignores the interlinkages between green and blue water. We used a sophisticated model to simulate the complex eco-hydrological processes by considering intensive human activities induced strong exchanges between surface water and groundwater, while traditional approaches using lumped or semi-distributed model might be insufficient to simulate all the necessary hydrological elements required by the analysis. We investigated green and blue water regime for different human and natural ecosystems in a detailed way and revealed the water competition dynamics between human and nature. These provide crucial implications for water management in an arid or semi-arid region and they are not shown by any other previous works.

**Comment 1**

The objectives of the paper state: "(1) How integrated hydrological modeling could efficiently and effectively simulate green and blue water dynamics while emphasizing the interlinkages between them; (2) How the implication of such green and blue water assessment could support basin-scale water resources management to address human-nature water conflicts." Both objectives are too generic. First, surely there are studies that have addressed these problems. The objectives of the paper need to reflect what this study does in addition. Second, this paper does not address these objectives in general. It is limited to a specific application in the Heine catchment. This needs to be reflected.

**Author's response**

Thank you very much for the comment. Regarding to the first objective "(1) How integrated hydrological modeling could efficiently and effectively simulate green and blue water dynamics while emphasizing the interlinkages between them;", we are sorry that we did not state it very clear here. Indeed there are many studies have been done on green and blue water resources assessment, as we have already mentioned in the introduction section. But studies that consider the complex interactions between green and blue water are rare. Even some of them integrated the interlinkages between green and blue water in the study by considering processes like irrigation and capillary rise, they still failed to provide a thorough analyzing regarding to explicit depiction of the dynamics, regimes and the corresponding implications for water management. In our work, the incoming and outgoing green water flow in the domain is not balanced. In

this region, dense irrigation system, including water pumping from wells and water diversion from rivers, was build here due to the limited precipitation here. Intensive irrigation activities affects the water cycling dramatically, especially causing strong surface water and groundwater exchanges. We want understand not only how much of green and blue water flow in and flow out of the domain, but also how green and blue water transformed between each other to match the imbalanced green and blue water flow regime. This is one of the main objectives of our study. The model we used is capable to simulate both surface hydrology and groundwater hydrology, as well as explicitly consider irrigation, water diversion and groundwater pumping. This allows us to explicitly investigate the green and blue water dynamics by emphasizing the interlinkages between them. Based on the Referee's suggestion, we modified our first objectives as following.

**Author's changes in manuscript**

Page 3, Line 28.

**original:** "(1) How integrated hydrological modeling could efficiently and effectively simulate green and blue water dynamics while emphasizing the interlinkages between them;"

**updated:** "(1) How integrated hydrological modeling could efficiently and effectively simulate green and blue water dynamics while emphasizing the interlinkages between them in an arid endorheic river basin by explicitly consider irrigation, water diversion and groundwater pumping;"

**Author's response**

Regarding to the second objective "(2) How the implication of such green and blue water assessment could support basin-scale water resources management to address human-nature water conflicts.", we agree on the Referee's comment that the specific application in an arid catchment should be reflected. We have now modified it as following.

**Author's changes in manuscript**

Page 3, Line 30.

**original:** "(2) How the implication of such green and blue water assessment could support basin-scale water resources management to address human-nature water conflicts."

**updated:** "(2) How the implication of such green and blue water assessment could support basin-scale water resources management to address human-nature water conflicts in an arid endorheic catchment."

**Comment 2**

Title is too generic. It could be a good title for a book or for a conference session. It would be appropriate if the paper was so comprehensive to address a problem that was never addressed and to a level that the problem is definitely solved. It is not the case of an application of a model on the Heihe basin.

**Author's response**

Thank you for the suggestion. We agree that the title is too generic. As already mentioned in Response to Comment 1, we would like to use a appropriate hydrological model to improve understanding the hydrological interactions between green and blue water which is usually ignored by other studies. However, the interlinkages between green and blue water play critical roles in the water cycling, especially in the region with intensive human activities. We believe such thorough study could provide crucial implications for water management in arid and semi-arid catchment, especially for the regions that have both human and natural ecosystems. Based on the Referee's suggestion, We changed the title to "Assessing Green and Blue Water in An Arid Catchment: Understanding Interactions and Making Balance between Human and Nature"

**Comment 3**

The introduction speaks about geological eras and problems of mankind but should be tailored to the specific advances that the paper wants to make.

**Author's response**

Indeed, we mentioned in the introduction that the humanity's impact is so profound now and this could lead to increased pressure, competition and conflict over freshwater resources use between human and nature. It is exactly the reason we do the research on an catchment with intensive human activities which might threaten the natural ecosystems. In this work, we also did the analysis on the water competition between human and natural ecosystems. Results show that the blue water resources are used by human with a higher priority. This implies that the natural ecosystems will face increased risk over freshwater use if competition increases. Therefore, the mention of Anthropocene in the introduction section is necessary for this work.

**Comment 4**

The model selection is a crucial methodological choice. A fully distributed hydrological model is selected for the following reasons: "(1) This study aims to assess the water resources by investigating the interlink between green water and blue water. The selected model is capable to simulate all the necessary hydrological elements for this analysis due to the capacity of the model for detailed depiction of interactions between groundwater and surface water. (2) Gridded hydrological simulations from distributed model are essential for spatial investigation on green and blue water." Both arguments are not convincing. Re argument 1, what are all the necessary hydrological elements needed to assess green and blue water components? I struggle to see why one would need to model everything if interested only in such partitioning. Re argument 2, why gridded simulations are essential? Can one go with a couple of HRUs?

**Author's response**

Thanks for the comment. Response to argument 1 issue. One of the objectives of this study is to investigate the interlinkages between green and blue water in an area with intensive irrigation, while the irrigation system consists of groundwater and surface water irrigation infrastructures. To achieve the goal, in addition to simulate general surface water and groundwater hydrological elements, such as runoff, infiltration, groundwater recharge and so on, we have to also simulate the entire irrigation system in the domain. These simulations includes pumping water from wells, surface water diversion from rivers, seepage flow from canals and return flow to the rivers or groundwater. All above-mentioned elements that are necessary for this work are referred to the necessary hydrological elements in this study. We did not, theoretically cannot, model everything. In conclusion, necessary hydrological elements refer to the basic requirements of data or simulation for hydrological problems solving. For instance, analyzing the water balance

in a basin without any human activities, the necessary hydrological elements for such study could be some general variables like precipitation, infiltration, evaporation and runoff.

Response to argument 2 issue. Indeed, spatial analysis can also be done with simulations on a couple of HRUs. We are sorry that we did not state it clear here. The advantage of grid-based model structure allows interaction between grid cells with a routing system. HRU-based model usually needs sub-basin delimitation and simulate for each sub-basins. Both of them allow for spatial investigation. However, grid-based model simulations could provide smooth information which can better reveal spatial heterogeneity. With the HRU-based (usually sub-basin scale) simulations, you will see gradients with suddenly changes. Based on the Referee's suggestion, we have modified our manuscript as following.

**Author's changes in manuscript**

Page 5, Line 2.

**original:** "(2) Gridded hydrological simulations from distributed model are essential for spatial investigation on green and blue water."

**updated:** "(2) Gridded hydrological simulations from distributed model are preferred for spatial investigation on green and blue water to reveal the spatial heterogeneity."

**Comment 5**

The calibration and validation of the model is deferred to another study. This limits the relative contribution of this paper to an analysis of already available results, and complicates the assessment of the appropriateness of methods.

**Author's response**

Thank you for the comments. We refer the calibration and validation to another study for the sake of avoiding the duplicate work that require high computing cost. Indeed, the model usually has to be calibrated and validated before application. The model we used is developed and well calibrated and validated by other study but in exactly the same regions. The same model setups, configurations and parameters are taken in our study. We believe it is fair to use it for another application without further calibration and validation in the same region. There are also other studies did it in the similar way. For instance, Döll et al. (2012) used the calibration parameter values of WaterGAP 2.1g to run the version 2.1h for another application. We are sorry that we did not state clearly about the calibration and validation issues in the manuscript. We thus changed the description of this part accordingly (see it in the following section: Author's changes in manuscript).

We did not take the already available results for the analysis of this work. As we mentioned in the manuscript (Section 2.4), the model is run at 1 km x 1 km spatial resolution at daily scale. We run the model with well calibrated set-ups and configurations in the same region with improved driving forces. The simulations are then used for green and blue water analysis.

**Author's changes in manuscript**

Page 6, Line 4.

**original:** "Since the improved GSFLOW model has already been well calibrated and validated in this region by Tian, Zheng, Zheng et al. (2015a) and Tian, Zheng, Wu et al. (2015b), it is directly run for the study area without any further parameter tuning."

**updated:** "The model has already been well calibrated and validated in this region by Tian, Zheng, Zheng et al. (2015a) and Tian, Zheng, Wu et al. (2015b). In this study, the same set-ups and configurations including calibrated parameters are taken for the model. Therefore, it is directly run for the study area without any further parameter tuning."

**Comment 6**

There is no uncertainty analysis in this paper and no comparison with alternative models. Considering that green and blue water are derived quantities form a calibration on other type of data (ie streamflow), I suspect that there is large uncertainty in the analysis. Perhaps something that is difficult to do with such a complex model, but perhaps also something that would speak in favour of different methodological choices? It would be also good to see how different would results be if applying a much simpler model, and if there are arguments to prefer the results of one model over the other.

**Author's response**

Thanks for comment. Indeed, we did not analysis the uncertainty in this work. The reasons are followed. (1) Usually, the model is not calibrated directly on the green or blue water flow itself, as the blue or green water consists of a few hydrological fluxes. E.g. blue water flow refers to the sum-up of the surface runoff, subsurface runoff and groundwater recharge. Therefore, even green and blue water are analyzed, the model is still calibrated and validated on the traditional hydrological fluxes. (2) The model is calibrated and validated with streamflow, ground water table and evapotranspiration rather than only one variable (figures from Tian et al. (2015) are shown in the following.). This ensures the accuracy of the simulations.

We agree that it would be good to see the comparison with alternative models. We also tried on this. However, the research domain we selected is installed with intensive irrigation systems. It is hard to find another models that can also simulate such complex irrigation system and irrigation activities. Indeed, there are some models have been applied in HRB for hydrological simulation, e.g. SWAT (Zang et al., 2012; Zang et al., 2015). But these studies are mainly focus on the climate change or land used induced impacts on hydrological cycle. None of them is able to capture the characteristics of such complex irrigation system,. The model we used (it is named as HEIFLOW in Li et al. (2018)) is so far the only model that can simulate the complex irrigation system in HRB. This is due to the reason that this model is tailored for HRB under the support of a key research program "Integrated research on the eco-hydrological process of the Heihe River Basin" which is launched by National Natural Science Foundation of China since 2010.

**Comment 7**

The treatment of the interception process is unclear. It appears that intercepted water has been incorporated in green water, but as already noted by other reviewers, this may be inappropriate.

[Figure]

Figure 1: Comparison of simulated and observed monthly streamflow in both calibration and validation periods: (a) Gaoai station; (b) Zhengyixia station; (c) Shaomaying station; and (d) Langxinshan station. (Tian et al., 2015)

[Figure]

Figure 2: Calibration and validation results for groundwater. (a) Annual average depth to water table (DW) for the calibration period (2001–2005); (b) annual average DW for the validation period (2006–2007); (c) seasonal variation of the groundwater level at a well in the domian. (Tian et al., 2015)

[Figure]

Figure 3: Comparison of monthly basin-wide ET simulations by the GSFLOW model and the ETWatch model. (Tian et al., 2015)

This comes back to clarifying what are the necessary processes to estimate green and blue water components (see point 4).

**Author's response**

Thank you for the comment. Indeed, we have originally incorporated the interception in green water as it is inherently a part of the green water. We also agree that the separation of interception from green water could improve the description of green and blue water flow regimes. We have addressed this issue already in the Response to RC1 Comment #1. We have also modified the text and figures in the manuscript accordingly. Please see the details in the Response to RC1 Comment #1.

**Comment 8**

Most conclusions are not related to the paper objectives

**Author's response**

Thank you for your comment. However, we cannot fully agree on this.

Our original objectives are "(1) How integrated hydrological modeling could efficiently and effectively simulate green and blue water dynamics while emphasizing the interlinkages between them; (2) How the implication of such green and blue water assessment could support basin-scale water resources management to address human-nature water conflicts."

Now the objectives are modified accordingly based on Referee's suggestions "(1) How integrated hydrological modeling could efficiently and effectively simulate green and blue water dynamics while emphasizing the interlinkages between them in an arid endorheic river basin by explicitly consider irrigation, water diversion and groundwater pumping; (2) How the implication of such green and blue water assessment could support basin-scale water resources management to address human-nature water conflicts in an arid endorheic catchment."

We here now list some of the conclusions.

"Even though the green water resources are the major resources in the research area - an arid river basin, the blue water resources from upstream are also crucial for the ecosystems in this region to meet the water demand. The transformation from blue water to green water plays a key role in the completed water cycling in this area as the water required for evaporation are extremely higher than the water stored in the root zone area (green water from precipitation)." This conclusion is based on the investigation of green and blue water dynamics and the interaction between them. All the data are simulated by an integrated hydrological model which is specified in the objective 1. Main results of this part shown in the Figure 3 in the manuscript. We believe it reflects the objective 1 of this work.

"Both the water availability and water consumption vary in time and space in the research area. Different hydrological processing mechanisms in ecosystems together with the spatial and temporal heterogeneity of water supply forms totally different green and blue water regimes in different ecosystems. The farmland ecosystem highly relies on the irrigation, while the forest relies on both the irrigation and capillary water. Both the grassland and desert ecosystems mainly rely on the green water from precipitation, while the desert ecosystem almost generates no runoffs." This conclusion is based on two key analyses. (1) the investigation of spatial and

temporal variability of the green and blue water; (2) the explicit analysis of green and blue flow regime in different ecosystems. All the analyses are highly rely on the integrated model simulations, especially the hydrological elements that reflect the interlinkages between green and blue water. The corresponding results are shown in Figure 4-6 in the manuscript. We believe this conclusion reflects the objective 1 of this work.

"The historical relationship between human water use and nature water use indicates that the blue water resources are used by human with a higher priority. Water consumption ratio of human increases with the decrease of the water availability. The natural ecosystems may take a higher pressure when the water competition between human and nature increases." This conclusion is based on the analysis of water consumption dynamics between human and nature. Such analysis reveals that the water used by human dramatically narrow down the water resources for nature. The corresponding results are shown in Figure 7 in the manuscript. We believe this conclusion reflects the objective 2 of this work.

**References**

Döll, P., Hoffmann-Dobrev, H., Portmann, F. T., Siebert, S., Eicker, A., Rodell, M., Strassberg, G. and Scanlon, B. R.: Impact of water withdrawals from groundwater and surface water on continental water storage variations, J. Geodyn., 59–60, 143–156, doi:10.1016/j.jog.2011.05.001, 2012.

Tian, Y., Zheng, Y., Zheng, C., Xiao, H., Fan, W., Zou, S., Wu, B., Yao, Y., Zhang, A. and Liu, J.: Exploring scale-dependent ecohydrological responses in a large endorheic river basin through integrated surface water-groundwater modeling, Water Resour. Res., 51(6), 4065–4085, 2015.

Zang, C., Liu, J., Velde, M. van der and Kraxner, F.: Assessment of spatial and temporal patterns of green and blue water flows under natural conditions in inland river basins in Northwest China, Hydrol. Earth Syst. Sci., 16(8), 2859–2870, 2012.

Zang, C., Liu, J., Gerten, D. and Jiang, L.: Influence of human activities and climate variability on green and blue water provision in the Heihe River Basin, NW China, J. Water Clim. Chang., 6(4), 800–815, 2015.

Li, X., Cheng, G., Ge, Y., Li, H., Han, F., Hu, X., Tian, W., Tian, Y., Pan, X., Nian, Y., Zhang, Y., Ran, Y., Zheng, Y., Gao, B., Yang, D., Zheng, C., Wang, X., Liu, S. and Cai, X.: Hydrological Cycle in the Heihe River Basin and Its Implication for Water Resource Management in Endorheic Basins, J. Geophys. Res. Atmos., n/a-n/a, doi:10.1002/2017JD027889, 2018.

---

## Author Comment (AC3) · 26 Jul 2018

We would like to thank S. Mylevaganam for his interest in this topic and for the valuable comments to improve our manuscript. Our point-by-point response to the comments is given in the following:

**General comments**

**Comment 1**

In the current version of the manuscript, the distinction between semi-distributed and fully distributed hydrological models is not apparent. What is meant by semi-distributed hydrological model? What is meant by fully distributed hydrological model? On what basis the selected model is considered as a fully distributed hydrological model (see P-4 LN-28)? Is the definition of fully distributed model meant for simulating the "necessary" hydrological element for the analysis of interest (see P-4 LN-31)? If the definition of fully distributed model is preserved only for the "necessary" elements, how would it lead to conclude that the selected model is a fully distributed hydrological model? Are there elements other than the necessary elements that are not fully described in the selected model? From the reader's point of view, having a fully distributed hydrological model is precluded with our understanding on the processes that define the hydrological system, even though many efforts have been made and are in progress to better understand the hydrological system. Moreover, from the reader's point of view, in no way, the selected model can represent a fully distributed hydrological model, considering the way the selected model has been conceptualized as described in the manuscript(see P-5 LN-1-15).

**Author's response**

Thank you for the comment. The semi-distributed model refers to the model which is able to simulates the spatial information, but it divides the catchment into sub-basins that in turn are divided into HRUs (hydrological response unites). Semi-distributed model lumps meteorological variables and physical parameters into sub-basins, thus, it is usually easy to setup and require shorter time relatively. The fully distributed model refers to the model that are capable of capturing the explicit spatial distribution of input variables, e.g. meteorological conditions, land use, soil characteristics and so on (Abu El-Nasr et al., 2005; Devia et al., 2015). Therefore, it can also provide explicit spatial simulations of hydrological conditions. It usually divided the catchment into regular pixels and simulate hydrological processes for each pixel and considers also the interaction between pixels. The fully distributed model is usually date intensive and needs more computation costs. The model we used in this study is a fully distributed mode that is able to reveal spatial heterogeneity in a detailed way.

**Comment 2**

The current version of the manuscript is based on a model that was calibrated and validated by Tian et al, 2015(https://www.sciencedirect.com/science/article/pii/S1364815214003016). Therefore, the research work that is carried out and presented in the current version of the manuscript solely depends on the calibrated and validated model. Moreover, around 82.1% of the LULC (see P-4 LN-14) of the domain of interest is desert. Therefore, it becomes paramount to evaluate the calibrated and validated model. Otherwise, the evaluation of this manuscript will be based on the assumption that the authors of that manuscript (https://www.sciencedirect.com/science/article/pii/S1364815214003016) are reliable and well-known in the field of hydrology, and/or the editorial board went through the calibrated and validated model carefully and ensured that the model is flawless. Moreover, the published manuscript (https://www.sciencedirect.com/science/article/pii/S1364815214003016)) is not feely available for a reader of an open-access journal (i.e., HESS) to evaluate this manuscript.

**Author's response**

Thank you for you comment. For the calibration and validation issue, we have explained in the Response to Comment 5 of Anonymous Referee # 2. For the sake of completeness, we have listed them in the following.

We refer the calibration and validation to another study for the sake of avoiding the duplicate work that require high computing cost. Indeed, the model usually has to be calibrated and validated before application. The model we used is developed and well calibrated and validated by other study but in exactly the same regions. The same model setups, configurations and parameters are taken in our study. We believe it is fair to use it for another application without further calibration and validation in the same region. There are also other studies did it in the similar way. For instance, Döll et al. (2012) used the the calibration parameter values of WaterGAP 2.1g to run the version 2.1h for another application.

We are sorry that the paper we referred is not free accessible. We have now included the critical results in the responses. We have also uploaded this paper and you can download it via the following link https://1drv.ms/f/s!AglvjnHO73u2geN-Cff-ikYApDVeZA

**Comment 3**

As per the "new" framework (see P-1 LN-11) implemented by the authors, the green and blue water resources are calculated for each pixel and then summed up (see P-6 LN-11). What is meant by this framework (fully distributed? new framework?)? How did the authors implement the streamflow routing? How did the authors account the pixels that represent the stream network? How did the authors calculate the surface runoff/excess rainfall at the uppermost point of a reach, which got routed along the reach? Moreover, what is the meaning of green water in the deserts (i.e., 82.1% of LULC)? Are these green water available in the root system for the plants in the deserts? If this is the case, the classification of LULC is misleading? Why are those areas considered as deserts?

**Author's response**

Thank you for the comment. The framework here means the proposed one for green and blue water assessment. It is not directly related to fully distributed issue. The main features of this proposed new framework consists of the following. (1) We investigated the blue and green water from both water supply and water consumption perspectives, while conventional studies focus only on one of them. (2) We considered the interaction between green and blue water in a detailed way, while conventional studies usually ignored the interactions.

Response to the questions "How did the authors implement the streamflow routing? How did the authors account the pixels that represent the stream network? How did the authors calculate the surface runoff/excess rainfall at the uppermost point of a reach, which got routed along the reach?". For streamflow routing, we used the original routing module named "Cascading-Flow Procedure" in GSFLOW model. The stream network is derived in the model based on the DEM and flow accumulation. Surface runoff and interflow are added to stream reaches by connecting HRUs to stream segments. The volume of runoff and interflow are distributed to each stream reach in a segment on the basis of the fraction of HRU associated with a stream

reach. For more details about such technique issues, we would like to refer to the manual of GSFLOW (Markstrom et. al., 2008).

Response to the green water in desert issue. The water evaporated in desert is also accounted as green water, even there is no plant there. This part of water is also known as the non-productive green water, while the water is used for transpiration refers to the productive green water (Rockström and Falkenmark, 2000). Sorry that we did not explain it clearly in the manuscript. We have now added the following explanation in the manuscript Page 2, Line 7 "The green water also consists of two components: The productive green water, i.e. the transpiration involved in biomass production in terrestrial ecosystems, and the non-productive green water, i.e. interception and soil evaporation (Rockström and Falkenmark, 2000)."

**Author's changes in manuscript**

Page 2, Line 7.

After "... aquifers and dams that can be extracted for human use (Falkenmark and Rockström, 2006)." the following sentence is inserted "The green water also consists of two components: The productive green water, i.e. the transpiration involved in biomass production in terrestrial ecosystems, and the non-productive green water, i.e. interception and soil evaporation (Rockström and Falkenmark, 2000)."

**Comment 4**

As per the authors, the study area (i.e., HRB) is impacted by "heavy human activities" and the hydrological cycle is "dramatically" altered. Moreover, the study area has "strong" GW and SW exchanges (see P-4 LN-5). Do these statements need supportive texts/references?

**Author's response**

Thank you for the comment. We have now included the supportive citation.

**Author's changes in manuscript**

Page 4, Line 5

Citation is added. "... HRB are impacted by heavy human activities and the hydrological cycling is dramatically altered (Zhou et al., 2014)"

Page 4, Line 6

Citation is added. "... HRB has strong groundwater and surface water exchanges which influences the interactions between green water and blue water (Zhu et al., 2008)"

**References**

Abu El-Nasr, A., Arnold, J. G., Feyen, J. and Berlamont, J.: Modelling the hydrology of a catchment using a distributed and a semi-distributed model, Hydrol. Process., 19(3), 573–587, doi:10.1002/hyp.5610, 2005.

Devia, G. K., Ganasri, B. P. and Dwarakish, G. S.: A Review on Hydrological Models, Aquat. Procedia, 4(Icwrcoe), 1001–1007, doi:10.1016/j.aqpro.2015.02.126, 2015.

Markstrom, S. L., Niswonger, R. G., Regan, R. S., Prudic, D. E. and Barlow, P. M.: GS-FLOW—Coupled Ground-Water and Surface-Water Flow Model Based on the Integration of the Precipitation-Runoff Modeling System (PRMS) and the Modular Ground-Water Flow Model (MODFLOW-2005), U.S. Geol. Surv., (Techniques and Methods 6-D1), 240, doi:10.13140/2.1.2741.9202, 2008.

Rockström, J. and Falkenmark, M.: Semiarid Crop Production from a Hydrolog- ical Perspective: Gap between Potential and Actual Yields, Critical Reviews in Plant Sciences, 19(4), 319-346, 2000.

Zhou, S., Huang, Y., Yu, B. and Wang, G.: Effects of human activities on the eco-environment in the middle Heihe River Basin based on an extended environmental Kuznets curve model, Ecol. Eng., 76, 14–26, doi:10.1016/j.ecoleng.2014.04.020, 2015.

Zhu, G. F., Su, Y. H. and Feng, Q.: The hydrochemical characteristics and evolution of groundwater and surface water in the Heihe River Basin, northwest China, Hydrogeol. J., 16(1), 167–182, doi:10.1007/s10040-007-0216-7, 2008.

---

## Author Comment (AC4) · 26 Jul 2018

We would like to thank J.F. Schyns for his interest in this topic and for the valuable comments to improve our manuscript. Our point-by-point response to the comments is given in the following:

**General comments**

The authors of this paper aim to improve the understanding of the hydrological interactions between green and blue water, and the relation between water for agriculture versus water for natural ecosystems. They study an arid endorheic river basin in China and use a coupled groundwater-surface water model. I have five major concerns with this manuscript:

**Comment 1**

The literature review on the one hand is very lengthy – going into many directions that seem not so relevant for this paper – while on the other hand key references are not included or not discussed properly. In the end it remains vague what the exact contribution of this paper is and where the novelty lies.

Perhaps the most relevant study which is not mentioned is the one by Weiskel et al. (2014). They used a detailed distributed water balance model to simulate green-blue water fluxes across the US and develop a classification of hydrological regimesunits based on these green-blue water fluxes. Another study that looked into the interaction between green and blue water fluxes is the one by Chukalla et al. (2015). They developed a method to separate cropland evaporation into green and blue fractions based on the ingoing and outgoing fluxes of the water balance. Other recent studies of relevance that could be included are: Lathuillière et al. (2018) and Xu & Wu (2018). The paper by Schyns et al. (2015) is referred to in an odd context in the manuscript (page 3, lines 21-23).

It is not clear what is the (novel) contribution of this manuscript. The conclusion section contains some claims on the novelty of the research which are strongly overstated: (a) "This study for the first time assesses the water resources by considering not only the blue and green water but also their interconnections."; and (b) "This study also investigated the blue and green water from both water supply and water consumption perspectives, while conventional studies focus only on one of them". Regarding statement (a), the studies by Weiskel et al. (2014) and Chukalla et al. (2015) considered this in a detailed manner. Basically, all studies that use a hydrological model or vegetation or crop growth model with a proper water balance in there take into account the interaction between green and blue water (e.g. Rost et al. (2008); Hanasaki et al. (2010)). Regarding statement (b), papers on combined green-blue water scarcity have studied green and blue water consumption versus green and blue water availability (Rockström et al., 2009; Gerten et al., 2011; Kummu et al., 2014). See also Schyns et al. (2015).

**Author's response**

Thank you for your comment. However, we cannot fully agree with the reviewer that the literature review goes into many directions and not relevant for the paper. All the studies we have mentioned strongly support the brief summary we have stated in Page 2, Line 24 "These green and blue water related researches generally can be categorized into two groups depend on different perspectives of the investigations: (1) Assessing green and blue water availability and their dynamics by using hydrological model or water-balance model. (2) Assessing green and blue water use or consumption by using the water resources model, agriculture model or dynamic vegetation model."

Regarding to citation of the relevant papers. Firstly, thank you for the nice paper introducing. As we have already explained in the last paragraph, the paper we have discussed are tailored to our introduction section and strongly supports the objectives of our study. It is hard for us to cite and discuss all the papers that related to green and blue water. They are too many and the papers that are cited and discussed should be tailored to the framework of this study. Indeed, the papers provided by the reviewer are relevant to the green and blue water analysis. However, they not the most relevant studies to our work that aims on a thorough green and blue water assessment. As already mentioned by the reviewer, Weiskel et al. (2014) simulated the green and blue water fluxes, but it focused on hydro-climatic regimes classification. Chukalla et al. (2015) separated the cropland evaporation into green and blue fractions, but it focused on water footprint analysis. Since the works from Weiskel et al. (2014) and Chukalla et al. (2015) still fit the context in Page 2, Line 22. We have cited these two papers in the end of the following sentence "In addition, many other studies have been done that are related to the green and blue water resources."

Regarding to the citation of "Schyns et al. (2015)", we are sorry for this mistake. We have now check again the citation relevance through the paper. The citation "Schyns et al. (2015)" has been moved from Page 3, Line 23 to Page 2, Line 10. As this paper emphasized the importance of green water.

Regarding to the conclusion issue (a) "This study for the first time assesses the water resources by considering not only the blue and green water but also their interconnections.", we have addressed this issue in the response to comments of Referee Prof. HHG Savenije ("Some small corrections:" suggested by Prof. HHG Savenije).

Regarding to the conclusion issue (b) "This study also investigated the blue and green water from both water supply and water consumption perspectives, while conventional studies focus only on one of them". Indeed, the studies on water scarcity usually compared the water availability and consumption. However, in our work, we reveals the imbalance between water availability and consumption and also reflect the transformation between green and blue water to reflect such imbalance. This is totally different to what the reviewer mentioned, i.e. using water consumption versus water availability to get a water scarcity index.

**Author's changes in manuscript**

Page 2, Line 22

Two more citations are added. "... In addition, many other studies have been done that are related to the green and blue water resources (Johansson et al., 2016; Mekonnen and Hoekstra, 2011; Rost et al., 2008; Sulser et al., 2010; Weiskel et al., 2014; Chukalla et al., 2015)."

Page 2, Line 10

A citations are added. "... However, green water plays a critical role in terrestrial ecosystem, especially in arid and semi-arid regions (Liu et al., 2009a; Rockström et al., 2007; Rost et al., 2008; Schyns et al., 2015)."

**Comment 2**

The definitions of green and blue water in this manuscript deviate from previous studies for unknown reasons, and the definitions are mutually inconsistent. Since the focus of the manuscript

is on the interactions between green and blue water flows (as put forward prominently in the title and introduction), this is a serious methodological flaw, which really makes me question the scientific quality of this work.

Various definitions of green water exist (see Schyns et al. (2015) section 2.3), though most studies define the green water flow as actual evapotranspiration (or more preferably called evaporation (Savenije, 2004)) from land, excluding the part of evaporation that is the result of blue water resources that have been redirected to the soil moisture through irrigation, capillary rise, or natural flooding. The authors have chosen their own definition of green water: "The green water resources from precipitation are calculated by summing up the infiltration simulated by the model for a certain period (e.g. annual scale), as the infiltrated water from precipitation will be stored in the unsaturated soil and eventually be used by the terrestrial ecosystems." This definition is incomplete and inconsistent with how blue water is defined. Water that infiltrates into the unsaturated zone of the soil will in part evaporate – through soil evaporation and through plant transpiration – and in part it will add to groundwater and surface water through percolation and interflow. Rockström and Falkenmark (2000) refer to this as the 'second partitioning point'. It is thus not true that the "infiltrated water from precipitation will be stored in the unsaturated soil and eventually be used by the terrestrial ecosystems" as the authors state. Infiltrated water will in part contribute to blue water resources. Furthermore, the authors' definition of green water does not include the intercepted rainwater that evaporates. Evaporation of intercepted rainwater is also part of the green water flow, albeit a non-productive vapour flow (Rockström and Falkenmark, 2000).

The authors have the following definition of blue water: "The blue water resources from precipitation are calculated by summing up the model simulated surface runoff, subsurface runoff and the groundwater recharge." Since infiltrated precipitation contributes to subsurface runoff and groundwater recharge as explained above, the used definitions of green and blue water are inconsistent and double-counting occurs.

The authors speak of irrigation and capillary rise as a transformation of blue to green water, and use the following definitions of green and blue water consumption: "The green water consumption refers to the evaporation in terrestrial pixels and the blue water consumption refers to the evaporation in open water pixels." I find this highly confusing, since it suggests that irrigation (and capillary rise) is accounted for by the authors as green water consumption, while previous studies all see this as blue water consumption (e.g. Oki & Kanae (2006); Rost et al. (2008); Liu & Yang (2009); Hoekstra & Mekonnen (2012); Hanasaki et al. (2010); Siebert and Döll (2010)). In fact, this means that this manuscript treats all agricultural water use as green water. An example from the paper: ". . .while the second highest green water consumption ecosystem is farmland (24.4%) partly due to the intensive irrigation" (page 7, lines 29-30). Moreover, the quoted definition suggests that open water evaporation is a form of blue water consumption, while in fact open water evaporation is purely natural, unless we are talking about open water evaporation of man-made reservoirs (Hogeboom et al. (2018)).

The definition of water availability that is put forward is also not clear: "The water availability in this study refers to the amount of received water resources for a certain period." What is meant by the received water resources? Simply precipitation?

**Author's response**

Thanks for the comment. In our work, we strictly follow the definition of green and blue water from the work of Falkenmark and Rockström, (2006) as we mentioned in the manuscript Page 2, Line 4. However, we are very sorry that we made a mistake on the description of green water

calculation (Page 6, Line 13), consequently causing confusion on blue water calculation. Thank you for pointing it out. Actually, for green water calculation we did not sum up the infiltration but the soil moisture recharge. Therefore, this is no double-counting issue on blue water. We have checked again through the manuscript and corrected accordingly now as following. "The green water resources from precipitation are calculated by summing up the soil moisture recharge simulated by the model for a certain period (e.g. annual scale), as the part of water will be stored in the unsaturated soil and eventually be used by the terrestrial ecosystems.".

Regarding to the "interception" issue, we do consider the interception as a part of green water. We have given a detailed explanation about this issue in the response to comments of Referee Prof. HHG Savenije (General comment # 1).

"The water availability in this study refers to the amount of received water resources for a certain period." The received water resources consists of precipitation and also the water from upstream.

**Comment 3**

The second objective of this manuscript is to study the relation between water for humans versus water for nature, as put forward in the introduction and the manuscript title. However, this is only addressed superficially without even mentioning the term 'environmental flow (requirements)' in the manuscript.

**Author's response**

Thank you for the comment. Here, we are more interested in green and blue water regime and water use dynamics between natural and human ecosystems. 'Environmental flow requirements' is usually essential for water scarcity related analysis, which considers it as a constrain condition.

**Comment 4**

None of the three major findings presented in the conclusions are new insights. Three major findings are presented in the conclusions. The first one basically says that irrigation is important, since in arid areas soil moisture stemming from precipitation is insufficient for agriculture. The second one confirms this and mentions that the green- blue partitioning depends on the land use. The third one says that natural ecosystems may be under pressure when human water demand increases, and when water availability decreases the ratio of water use to availability increases (if demand remains the same). I fail to see what is new about these insights.

**Author's response**

Thank you for the comment. However, we cannot fully agree with the reviewer that None of the three major findings presented in the conclusions are new insights.

Response to the first major finding in the conclusion. Most of the studies in arid and semi-arid area emphasize the importance of the green water, as it is the major resources in such area. Our work did a thorough analysis on green and blue water flow regime and dynamics between them. Results show that in addition to green water, blue water also plays critical role in water cycling by investigating the transformation between blue and green water. This is not shown in any other previous works. We here emphasized the importance of the transformation from blue water to green water rather than the importance of irrigation.

Response to the second major finding in the conclusion. This conclusion is based the analysis on the green and blue water regime in different ecosystems. We have shown a very detailed green and blue water flow regime for different ecosystems. This is also not shown by any other previous works. We want to emphasize the importance of green and blue water regime differences which could provide crucial information for water management. This is totally different to what have mentioned by the reviewer, i.e. "the green- blue partitioning depends on the land use".

Response to the third major finding in the conclusion. We reveals the water use dynamics between human and nature in the research region. We emphasized that human uses water resources with a higher priority, this could pose higher risk on nature under changing condition, e.g. if the water competition increases. This is also not shown by any other previous works. The reviewer's comment "The third one says that natural ecosystems may be under pressure when human water demand increases, and when water availability decreases the ratio of water use to availability increases (if demand remains the same)" is really a misunderstanding.

**Comment 5**

The overall writing style is not on par with the level of a high quality paper, as indicated by the above examples of overstatements and definitions that are not fully clear. Also many (vague) claims are made without proper justification. Some examples from the conclusions section: "It allows us to explicitly assess the green and blue water resources beyond the water balance, while the traditional methods using lumped or semi-distributed model might be insufficient."; "Such sophisticated research framework allows us to take into consideration of all the important factors into water resources assessment as possible."; "The detailed analyses of green and blue water dynamics bring us a step further to understand the human and nature water use dynamics."; "It provides essential implications for water management under the changing environment that aims to make the balance between humankind and nature and towards sustainable development."

**Author's response**

Thank you for your comments that improved our manuscript a lot. We know that our manuscript is not perfect, we are trying our best to improve it. As we have explained in the last response. All the conclusion are based on what we have analyzed. We cannot agree with the reviewer that our conclusion are made without proper justification.